# A Numerical Reassessment of the Gulf of Mexico Carbon System in Connection with the Mississippi River and Global Ocean

Le Zhang[1] and Z. George Xue [1,2,3]

[1]Department of Oceanography and Coastal Sciences, Louisiana State University, Baton Rouge, LA 70803
[2]Center for Computation and Technology, Louisiana State University, Baton Rouge, LA 70803
[3]Coastal Studies Institute, Louisiana State University, Baton Rouge, LA 70803

*Correspondence to*: Z. George Xue (zxue@lsu.edu)

**Abstract.** Coupled physical-biogeochemical models can fill the spatial and temporal gap in ocean carbon observations. Challenges of applying a coupled physical-biogeochemical model in the regional ocean include the reasonable prescription of carbon model boundary conditions, lack of in situ observations, and the oversimplification of certain biogeochemical processes. In this study, we applied a coupled physical-biogeochemical model (Regional Ocean Modelling System, ROMS) to the Gulf of Mexico (GoM) and achieved an unprecedented 20-year high-resolution (5 km, 1/22°) hindcast covering the period of 2000 to 2019. The biogeochemical model incorporated the dynamics of dissolved organic carbon (DOC) pools and the formation and dissolution of carbonate minerals. The biogeochemical boundaries were interpolated from NCAR's CESM2-WACCM-FV2 solution after a comprehensive evaluation of 17 Global Climate Model (GCMs) products against available observations and global climatology products. Model outputs included generally interested carbon system variables, such as $pCO_2$, $pH$, aragonite saturation state ($\Omega_{Arag}$), calcite saturation state ($\Omega_{Calc}$), $CO_2$ air-sea flux, carbon burial rate, etc. The model's robustness is evaluated via extensive model-data comparison against buoy, remote sensing-based Machine Learning (ML) products, and ship-based measurements. Model results reveal that the GoM water has been experiencing an ~ 0.0016 yr$^{-1}$ decrease in surface $pH$ over the past two decades, accompanied by a ~ 1.66 µatm yr$^{-1}$ increase in sea surface $pCO_2$. The air-sea $CO_2$ exchange estimation confirms with several previous models and ocean surface $pCO_2$ observations that the river-dominated northern GoM (NGoM) is a substantial carbon sink, and the open GoM is a carbon source during summer and a carbon sink for the rest of the year. Sensitivity experiments are conducted to evaluate the impacts of river inputs and the global ocean via model boundaries. The NGoM carbon system is directly modified by the enormous carbon inputs (~15.5 Tg C /yr DIC and ~2.3 Tg C/yr DOC) from the Mississippi-Atchafalaya River System (MARS). Additionally, nutrient-stimulated biological activities create a ~105 times higher particulate organic matter burial rate in NGoM sediment than in the case without river-delivered nutrients. The carbon system condition of the open ocean is driven by inputs from the Caribbean Sea via Yucatan Channel and is affected more by thermal effects than biological factors.

## 1 Introduction

Carbon dioxide ($CO_2$) concentration in the atmosphere has increased approximately 150% from 1750 to 2019 (Le Quéré et al., 2018), and the storage and transport of carbon in Earth's ecosystem under the context of climate change has been receiving incremental attention over the past decades (Anav et al., 2013; Lindsay et al., 2014; Jones et al., 2016). The direction and magnitude of ocean-atmosphere $CO_2$ fluxes are subject to change with increasing atmospheric $CO_2$ concentrations (Smith & Hollibaugh 1993, Wollast & Mackenzie 1989), incremental ocean dissolved inorganic carbon (DIC) level (Torres et al., 2011), modification of the coastal alkalinity generation process (Renforth & Henderson, 2017), changes in organic matter (OM) remineralization patterns (Buesseler et al., 2020), river inputs (Yao & Hu, 2017), etc. As an enormous reservoir, the ocean has uptake some $170 \pm 20$ PgC (Le Quéré et al., 2018) since the industrial revolution. This alleviates the $CO_2$ accumulation rate in the atmosphere while inducing a consequent increase in ocean carbon level and a decrease in ocean $pH$ and calcium mineral saturation state ($\Omega$, Doney et al., 2009). Given the stake it holds in shaping climate feedback in the long term and the risk for coastal ecosystems under acidification stress, carbon sink quantities and their trends have been studied and monitored by multiple studies (Maher & Eyre, 2012; Czerny et al., 2013; Najjar et al., 2018; Bushinsky et al. 2019).

Nevertheless, mismatches in carbon flux estimates among different studies and difficulties in describing the spatial and temporal pattern of $pCO_2$ data collected from ship-based measurements left many vital questions unanswered. Global Earth System Models (ESMs) are essential tools for studying the linkage between the ocean carbon cycle and climate change. Extensive utilization of ESMs in hindcasting and coupled biogeochemistry provide pivotal information for understanding the carbon cycle on a global scale (Anav et al., 2013; Laurent, Fennel, & Kuhn, 2021; Lindsay et al., 2014; Jones et al., 2016; Todd-Brown et al., 2014). However, their relatively coarse spatial resolution is likely not appropriate to be directly compared with field measurements. It is imperative to apply high-resolution regional ocean models to understand carbon exchange and carbon budget at a regional scale. While high-resolution regional models have been developed to represent the complex patterns of ocean circulation and elemental fluxes on the continental shelves, the regional ocean carbon system is challenging to model and predict due to its high sensitivity to the boundary and initial conditions, uncertainties in the carbon pathway, and complex interactions between the atmosphere, ocean, and land (Hofmann et al., 2011).

The Gulf of Mexico (GoM) is a semi-closed marginal sea. The presence of the Mississippi-Atchafalaya River System (MARS) and the obstructions from Florida Strait and Yucatan Channel mitigate the impact of the global ocean on the GoM regarding water acidity and carbon levels. Allochthonous nutrients from river input, upwelling, and boundary shape the general pattern of the carbon cycling in the GoM (Cai et al., 2011; Chen et al., 2000; Delgado et al.,2019; Dzwonkowski et al., 2018; Laurent et al., 2017; Jiang et al., 2019; Sunda & Cai, 2012; Wang et al., 2016), and need to be properly included in the carbon system modeling in the GoM. Fennel et al. (2011) performed a coupled physical-biological modeling of the northern GoM (NGoM) shelf with the nitrogen cycle to describe the phytoplankton variability under the influence of the MARS covering the period

of 1990 to 2004. They found that biomass accumulation in the light-limited plume region near the Mississippi River delta was not primarily controlled bottom-up by nutrient stimulation because of the lack of nutrient limitation in the eutrophic zone. Xue et al. (2016) achieved a first GoM carbon budget and concluded that the export of carbon out of the Gulf via Loop Current is largely balanced by river inputs and influx from the air. Their regional carbon model used three sets of initial and open boundary conditions derived from empirical salinity-temperature-DIC-alkalinity relationships. Although this method of carbon system prescription leveraged the convenience of widely available physical variables and was feasible for regions with scarce DIC and alkalinity data, its reliability was questionable as temperature and salinity alone cannot fully describe the spatial and temporal pattern of these carbon variables. Laurent et al. (2017) presented a regional model study of the eutrophication-driven acidification and simulated the recurring development of an extended acidified bottom waters in summer on the NGoM shelf. They proved that the acidified waters were confined to a thin bottom boundary layer where the production of $CO_2$ was dominated by benthic metabolic processes. Despite reduced $\Omega$ values being produced at the bottom due to acidification, these regions remain supersaturated with aragonite. Chen et al. (2019) presented a unified model to estimate surface $pCO_2$ by applying Machine Learning (ML) methods to remote sensing data and cruise $pCO_2$ measurements. Their ML model confirmed that the GoM was a carbon sink. Recently Gomez et al. (2020) performed another GoM carbon model study covering the period of 1981 to 2014. Their model initial and boundary conditions were derived from a downscaled Coupled Model Intercomparison Project 5 (CMIP5) Modular Ocean Model (25km resolution, Liu et al., 2015). Their model results showed that GoM was a sink for atmospheric $CO_2$ during winter-spring, and a source during summer-fall, producing a basin-wide mean $CO_2$ uptake of 0.35 mol $m^{-2}$ $yr^{-1}$. Nevertheless, their model does not include the DOC pool or the calcification process, which are imperative to describe the dynamic of DIC and alkalinity in the ocean.

Despite the above carbon system regional modeling efforts, we notice that several processes that could contribute significantly to the carbon cycle in the GoM have not been investigated yet. The carbon cycle in the ocean is linked with the nutrient cycle through photosynthetic activities, calcification, and OM remineralization (Anav et al., 2013; Farmer et al., 2021; King et al., 2015). OM remineralization could be the most critical mechanism regulating the ocean carbon system, followed by the $CaCO_3$ cycle (Lauvset et al., 2020), with the remineralization of small detritus accounting for over 40% of the DIC production on the shelf (Laurent et al., 2017). Autochthonous nutrients from direct remineralization of OM determine the gradient of DIC in the euphotic layer (Boscolo-Galazzo et al., 2021; Boyd et al., 2019). During this process, the fast-sinking of OM and higher particulate to dissolved ratio foster a larger sedimentation rate and more significant DIC removal of the euphotic layers; on the contrary, slower sinking and faster decomposition rate of OM favors nutrient and DIC retention in the euphotic layers (Davis et al., 2019; Mari et al., 2017; Turner, 2015). The remineralization of land-derived OM and $CaCO_3$ precipitation are significant factors controlling air-sea $CO_2$ flux (Mackenzie et al. 2004). Studies have revealed the formation of marine $CaCO_3$ (Burton and Walter,1987; Inskeep and Bloom, 1985; Zhong and Mucci, 1989; Zuddas and Mucci, 1998) and the dissolution of marine $CaCO_3$ mineral is $\Omega$-dependent as well (Adkins et al. 2021). The $\Omega$ will be depressed with more $CO_2$ dissolves in seawater and can be used as an indicator for the buffering capacity of the ocean carbonate system. Given that $\Omega$ influences the calcification

rate of marine organisms and regulates the acidity of bottom waters, it should be considered in the $CaCO_3$ cycle for a comprehensive carbon cycle assessment.

By using the biogeochemical boundaries from one of the latest CMIP6 products, our model inherited the climate perturbation signals (Liao et al., 2020) and the accumulative effect of carbon variables from the global solutions. Our regional model includes critical carbon cycle processes lacking in previous efforts, including the most up-to-date carbonate chemistry thermodynamic parameterization, phosphate cycling, formation & dissolution of $CaCO_3$, and the inclusion of the DOC as a semi-labile carbon pool. The objective of this study is 1) to assess the feasibility and robustness of utilizing global model

products to drive a regional coupled physical-biogeochemical model, and 2) to examine the temporal trend of key variables of the carbon system ($pCO_2$, $pH$, air-sea $CO_2$ exchange, and $\Omega$) of the surface ocean in the GoM. In addition, to evaluate the impact of MARS and the global ocean on GoM's carbon cycling, two perturbed experiments are designed. The following sections are organized: model setup is given in Sect. 2; in Sect. 3, we validate the model's performance against buoy, remote sensing-based ML solution, and ship-based measurements; the trend of key carbon system variables over the past two decades

and an assessment of the contribution of riverine inputs and the global ocean are presented in Sect. 4; An evaluation of our regional model's performance against GCMs, climatology products, and existing regional models is given in Sect. 5, together with an outlook on future model development.

**2 Method**

**2.2 Model setup**

Our model is built on the platform of Coupled Ocean-Atmosphere-Wave-Sediment Transport modeling system (COAWST; Warner et al., 2010). COAWST is an open-source community model which includes the Model Coupling Toolkit to allow data exchange among three state-of-the-art numerical models: Regional Ocean Modelling System [ROMS, svn 820, Haidvogel et al., 2008; Shchepetkin and McWilliams, 2005], the Weather Research and Forecasting model [WRF, Skamarock, et al., 2005], and the Simulating Waves Nearshore model [SWAN, Booij et al., 1999]. The carbon model presented in this study is based on

a well-validated coupled physical-biogeochemical model by Zang et al. (2019 and 2020), which covers the entire GoM waters (Gulf-COAWST, Fig. 1). The Gulf-COAWST has a horizontal grid resolution of ~5 km and 36 sigma-coordinate (terrain-following) vertical levels. A third-order upstream horizontal advection and fourth-order centered vertical advection are used for momentum and tracer advection. The biogeochemical model is developed based mainly on the pelagic N-based biogeochemical model Pacific Ecosystem Model for Understanding Regional Oceanography (NEMURO, Kishi, et al., 2007;

Kishi et al., 2011). In this study, we extend from the original 11 state variables of the NEMURO, including nutrients ($Si(OH)_4$, $NO_3$, $NH_4$), plankton groups (ZP: predator zooplankton, ZL: large zooplankton, ZS: small zooplankton, PL: large phytoplankton, PS: small phytoplankton), dissolved organic nitrogen (DON), particulate organic nitrogen (PON), and opal (OPL) to 17, with added variables of phosphate ($PO_4$), particulate inorganic carbon (CalC), dissolved organic carbon (DOC),

oxygen ($O_2$), dissolved inorganic carbon (DIC) and total alkalinity (TA). The stoichiometry between carbon and nitrogen in the OM production and remineralization is set to 6.625 following Fennel (2008).

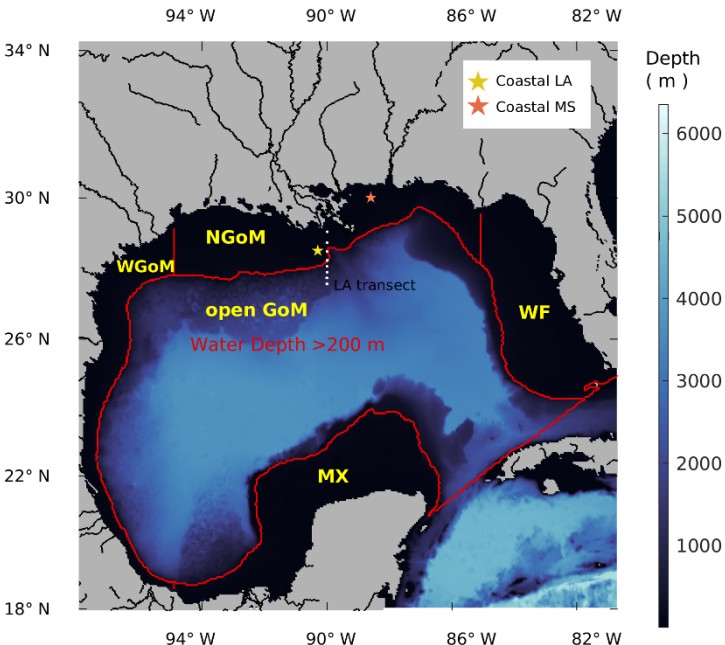

**Figure 1: Gulf-COAWST model domain with water depth in color (unit: m). Subregional definitions followed Xue et al. (2016), which are Mexico Shelf (MX), Western Gulf of Mexico Shelf (WGoM), Northern Gulf of Mexico Shelf (NGoM), West Florida Shelf (WF), and Open GoM.**

The revised biogeochemical model incorporates key processes regulating the carbon model, including primary production, river DIC, PON and DOC delivery, sediment carbon burial, $CO_2$ air-sea flux, $CaCO_3$ cycling, and OM remineralization (Fig. 2). Widely used carbon system variables, such as $pCO_2$, $pH$, $\Omega$, etc., are used as carbon system state indicators. The carbon module that takes in DIC, TA, $PO_4$, Si (dissolved inorganic silicon), salt, temp, for calculating $pCO_2$, $pH$, $\Omega_{Arag}$, and $\Omega_{Calc}$, largely followed the recommended best practices (Dickson, Sabine, & Christian, 2007; Eyring et al., 2016; Orr et al., 2017; Zeebe and wolf-Gladrow, 2001), with an updated parameter prescription for dissociation constants for carbonic acid ($K1$) and bicarbonate ion ($K2$) (Millero, 2010), and solubility products for aragonite $K_A$ and calcite $K_C$ (Mucci, 1983) with pressure effect (Millero, 1982; Millero, 2007).

Inorganic carbonate mineral (mainly $CaCO_3$) forms during the photosynthetic activities of some phytoplankton species and fosters aggregation of detritus and their sinking. The rate of $CaCO_3$ production followed a dynamic ratio regarding the primary production of small phytoplankton with low-temperature inhibition and enhancement during bloom conditions (Moore et al., 2004). The production and dissolution of $CaCO_3$ are important processes for ocean acidity regulation, as its production (by 1

unit) nominally takes away a unit of $[CO_3^{2-}]$ from water, which reduces the alkalinity and DIC by 2 and 1 unit, respectively. This process routinely happens during photosynthetic activities of some phytoplankton species (such as coccolithophores, parameterized implicitly as a portion of small phytoplankton in this model) and other marine calcifiers. Carbonate minerals produced in the euphotic zone could be treated as equivalent storage of alkalinity and are usually transported towards the ocean sediment through sinking. Aragonite and calcite are two common mineral phases of $CaCO_3$ secured by marine organisms and

are included in the model. $\Omega_{Calc}$ and $\Omega_{Arag}$ are calculated as the equilibrium product of $Ca^{2+}$ and $CO_3^{2-}$. When $\Omega > 1$, calcification is thermodynamically favored, and when $\Omega < 1$, dissolution is thermodynamically favored. In Eq. (1), $[Ca^{2+}]$ and $[CO_3^{2-}]$ are the concentrations of calcium and carbonate ions, respectively. $[Ca^{2+}]$ is determined through salinity (Millero, 1982; Millero, 1995), and $[CO_3^{2-}]$ is calculated through the carbon module. $K_{sp}$ is the stoichiometric solubility product dependent on pressure, temperature, and salinity. $K_{sp}$ is defined for aragonite and calcite as $K_A$ and calcite $K_C$, respectively.

$$\Omega = \frac{[Ca^{2+}][CO_3^{2-}]}{K_{sp}} \quad (1)$$

In our model, the sediment pool of sinking particles is a simplified representation of burial and benthic remineralization processes, where the flux of sinking materials out of the bottommost grid point is added to the sediment pool and enters the burial pool (remains inactive) with a dynamic ratio, the active sediment pools undergo enzyme-aided decomposition at rates

regulated by temperature and oxygen, and release corresponding influx of ammonium, DIC, and alkalinity at the sediment/water interface. Our model uses a $CO_2$ production ratio of 0.138 between sediment aerobic respiration and denitrification (Fennel et al., 2006) and an alkalinity production ratio of 1.93 between pyrite burial and denitrification (Hu & Cai, 2011). Upon sunk to acidified regions, the dissolution of $CaCO_3$, regulated by $\Omega$, can consume dissolved $CO_2$ and neutralize the acid.

The bulk formula for air-sea gas exchange for $CO_2$ is used following Wanninkhof (1992). Air-sea $CO_2$ flux is calculated with a timestep of 240 s and output in the form of a daily average. The gas transfer velocity coefficient of 0.31 cm h$^{-1}$ is used in Eq. (2).

$$F_{CO2} = k_{660} \left(\frac{Sc}{660}\right)^{-1/2} s \, \Delta pCO_2 \quad (2)$$

Where $F_{CO2}$ is the air-sea $CO_2$ flux in the unit of mmol $CO_2$ m$^{-2}$ d$^{-1}$. $Sc$ is the Schmidt number (nondimensional) (calculated following Wanninkhof, 2014), $s$ is the solubility of $CO_2$ in seawater in mol $CO_2$ m$^{-3}$ µatm$^{-1}$ (calculated following Weiss, 1974), and $\Delta pCO_2$ is the air-sea $pCO_2$ difference in µatm. The term $k_{660}$ is the quadratic gas transfer coefficient in cm h$^{-1}$ (converted to m d$^{-1}$). We calculated the air-sea $CO_2$ flux using the relationships of gas exchange with wind speed at 10 m over the sea

surface ($U_{10}$), following Wanninkhof (1992). We used the ocean convention for the $CO_2$ flux, i.e., a positive flux is defined as the ocean being a sink of atmospheric $CO_2$. Air $pCO_2$ level is prescribed using a fitted curve from column-averaged dry-air

mole fraction of atmospheric carbon dioxide from 2002 to the present derived from satellite product (merged dataset from SCIAMACHY/ENVISAT, TANSO-FTS/GOSAT, and OCO-2 [https://cds.climate.copernicus.eu/; Dils et al., 2014]).

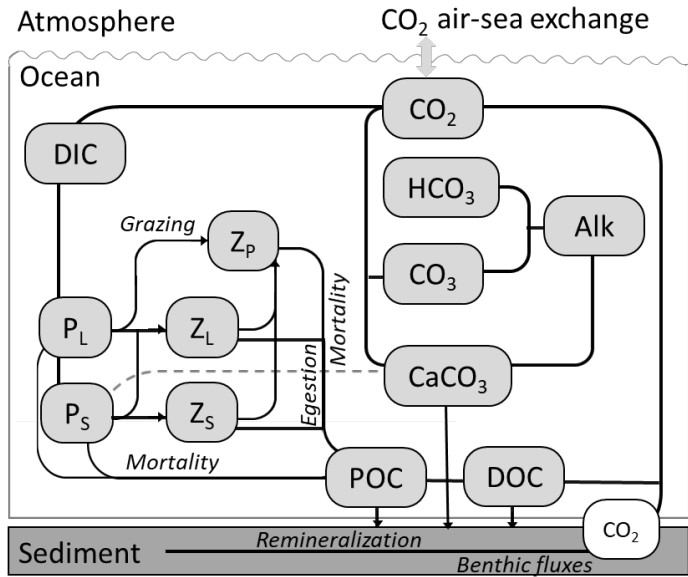

**Figure 2: Schematic plot showing major processes incorporated in the carbon cycle.**

We performed a 20-yr model hindcast covering the period of 1 January 2000 to 31 December 2019. The physical model setup was similar to that of Zang et al. (2020), with ocean physical initial and boundary conditions interpolated from the 1/12° data assimilated Hybrid Coordinate Ocean Model (HYCOM/NCODA, GLBu0.08/expt_19.1, expt_90.9, expt_91.0, expt_91.1, expt_91.2, and expt_93.0 [https://www.hycom.org; Chassignet et al. 2003]). Physical boundary conditions are of daily frequency and include u, v, ubar, vbar, zeta, temp, and salt. Atmospheric forcings with 6-hourly frequency include ground level or sea surface downwelling shortwave/longwave radiation, ground-level or sea surface upwelling shortwave/longwave radiation, surface air pressure, surface air temperature, relative humidity, precipitation, wind at 10 m were extracted from the NCEP Climate Forecast System Reanalysis (CFSR) (Saha et al. 2010) and Climate Forecast System Version 2 (CFSv2) (Saha et al. 2011). See Table A2 for a list of model forcing frequencies.

The Coupled Model Intercomparison Project 6 (CMIP6) participating GCMs consume enormous research resources and generate unprecedented knowledge on global carbon system evolution with a whole-ecosystem conservation perspective. Utilizing GCMs results in a refined regional model extends their research value, especially in bridging coarse GCMs product with *in situ* field observations. With the interannual variation estimated by GCMs, the regional model could take advantage of global models by using dynamic boundaries that reflect climate oscillations and carbon accumulation in oceanic waters. In this study, we first carefully evaluate various GCM products as candidates for initial and boundary conditions for the biogeochemical model. The choice of CESM2-WACCM-FV2, among other GCMs, is primarily based on its horizontal

resolution in the GoM region and its availability of nutrients and carbon variables (see Table A1 for more details). Monthly boundary conditions of the biogeochemical variables (DIC, DOC, TA, $NO_3$, $PO_4$, Si, $NH_3$ are extracted from CESM2-WACCM-FV2 solutions (historical, r1i1p1f1, nominal resolution 100 km, [Danabasoglu, 2019b]). As the global model simulation ended in December 2014, the biogeochemical boundary condition of 2014 was used repeatedly for the period from 2015 to 2019. The oxygen boundary condition is static without temporal changes since $O_2$ is not available from the CESM2-WACCM-FV2 and is interpolated from the World Ocean Atlas 2018 (WOA18) product (Boyer et al., 2018; García et al., 2019). Freshwater and terrestrial nutrient inputs from 47 major rivers discharged to the GoM are applied as point sources with daily frequency. River discharge and water quality data of rivers in the U.S. are collected from US Geological Survey (USGS) stations (https://maps.waterdata.usgs.gov). River DOC is prescribed following the values reported by Shen et al. (2012), with additional references from several other studies (Reiman & Xu, 2019; Stackpoole et al., 2017; Wang et al., 2013; Xu & DelDuco, 2017). Mexican river discharge data are collected from BANDAS (https://www.gob.mx/conagua). Water quality data of Mexican rivers is prescribed as the average of that of the Mississippi and Atchafalaya rivers. River nutrient and carbon load are reconstructed from available USGS observations (see Fig. 3 for time series of river DIC, TA input). Missing river alkalinity values are interpolated from climatological values, and missing river DIC values are calculated from *pH* and alkalinity using the MATLAB program CO2SYS (Lewis & Wallace, 1998). Validations of the model's performance in physics, nutrient cycle, and primary production can be found in Zang et al., 2019 and 2020. In this study, we focus on the model's performance in the carbon cycle, which is presented in the next section.

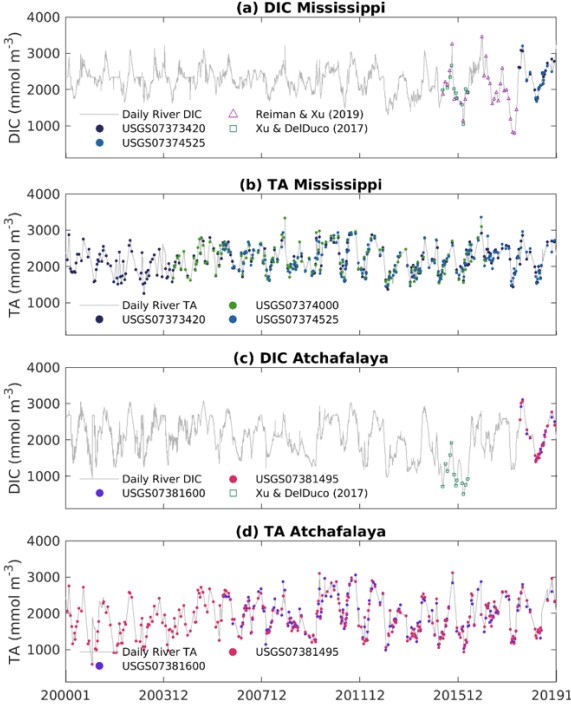

**Figure 3: River DIC, TA concentration prescribed in the model. Grey lines are the interpolated daily concentration values; colored data points are raw data collected from multiple sources.**

Since the model simulated DIC concentration in the water column is sensitive to initial conditions (Hofmann et al., 2011; Xue
et al., 2016), using the initial condition from a snapshot (January 2000) of the global model result would be appropriate as the
global model has been well stabilized up to the year 2000 from its "pre-industry" experiment. The regional model has the
benefit of swift spin-up, with the biogeochemical model typically completing its spin-up in one year (e.g., Große, Fennel,
Laurent, 2019; Laurent and Fennel, 2019; Laurent et al., 2021). We conducted a series of sensitivity tests and confirmed that
a one-year spin-up period (the year 2000) is sufficient for our current model setup. All results presented below are based on
model outputs from 2001 to 2019 unless otherwise specified. To quantify the impact of river discharge and the global ocean
on the carbon system in the GoM, in addition to the control experiment where the historical product of the CESM2-WACCM-
FV2 experiment is applied as the boundary conditions (from here, experiment "His"), two perturbed experiments, "Bry" and
"NoR" are added. The Bry experiment has clamped DIC and TA conditions as that of the year 2000 for all following years
while keeping all other experiment setups the same as that of the His. The NoR experiment eliminates the presence of all rivers
in the model while keeping the rest of the experiment set up the same as that of the His. As most available observations are
confined to the surface ocean, except for the GOMECC transects, for this study we focus on the surface ocean carbon condition
in the NGoM and Open GoM waters.

## 3 Validation

This section focus on the validation of the model results via comparison against autonomous mooring systems with surface
$pCO_2$ measurements, ship-based measurements from the Gulf of Mexico Ecosystems and Carbon Cruise transects (GOMECC,
Barbero et al., 2019; Wanninkhof et al., 2013; Wanninkhof et al., 2016), and $pCO_2$ underway measurements (data downloaded
from https://www.ncei.noaa.gov/access/oads/). Direct observations of the GoM carbon system have been recognized as
unbalanced among seasons due to fewer data points available in winter compared to other seasons. To overcome the sporadic
direct measurement dataset, we also performed a model-data comparison against the remote sensing-based ML product of sea
surface $pCO_2$ by Chen et al. (2019).

### 3.1 Model–buoy comparisons

Temporal variability of sea surface $pCO_2$ was recorded by the autonomous mooring system at two sites (CoastalMS and coastal
LA) operated by the Atlantic Oceanographic and Meteorological Laboratory (AOML) of the National Oceanic and
Atmospheric Administration (NOAA). The CoastalMS buoy site (location see Fig. 1, data coverage: 2009-01-14 to 2009-12-
09; 2011-03-17 to 2012-08-04; 2013-07-10 to 2014-02-10; 2014-02-10 to 2014-05-03; 2014-12-12 to 2015-03-22; 2015-03-
30 to 2016-09-22 2016-09-23 to 2017-05-29) is predominately impacted by the Mississippi River followed by the coastal
ocean, whereas the CoastalLA buoy site (data coverage: 2017-07-14 to 2017-11-07; 2017-12-14 to 2019-04-26; 2019-06-04
to 2020-08-12; 2020-08-12 to 2021-08-25; 2021-08-25 to 2021-11-29) is mutually influenced by the Mississippi River and the

coastal ocean. The high-frequency measurements provide a time-resolved picture of year-round changing $pCO_2$ values.

Temperature and salinity can influence the chemical equilibrium in the carbonate system, therefore shifting the $pCO_2$ values. Validating the temperature and salinity at these two mooring sites is a prerequisite before looking into the surface $pCO_2$ levels. In Fig. 4, the top four panels compare the sea surface temperature (SST) and salinity (SSS) between model and buoy measurements and show satisfying model-data agreements, with correlation coefficients larger than 0.75. At CoastalMS, the range for sea surface $pCO_2$ is 150~600 µatm. Sea surface $pCO_2$ records are more volatile at CoastalLA with a maximum value

> 800 µatm and a minimum value around 150 µatm. Following the salinity drop, $pCO_2$ at the CoastalMS site is simultaneously reduced, demonstrating the river's influence on both salinity and $pCO_2$. At CoastalLA, however, the $pCO_2$ level does not necessarily follow the trend of salinity, implying complex controlling factors in addition to the river inputs. The bottom two panels of Fig. 4 show an acceptable agreement between measured and simulated sea surface $pCO_2$, with a correlation coefficient of 0.27 between modeled and observed surface $pCO_2$ at the CoastalMS buoy location and a correlation coefficient

of 0.55 between modeled and observed surface $pCO_2$ at the CoastalLA buoy location. We notice model-data discrepancies in April 2018-04 at CoastalLA and July 2011 at CoastalMS and ascribe such bias to the uncertainty in the riverine DIC inputs prescription and the limited model horizontal resolution (~5 km).

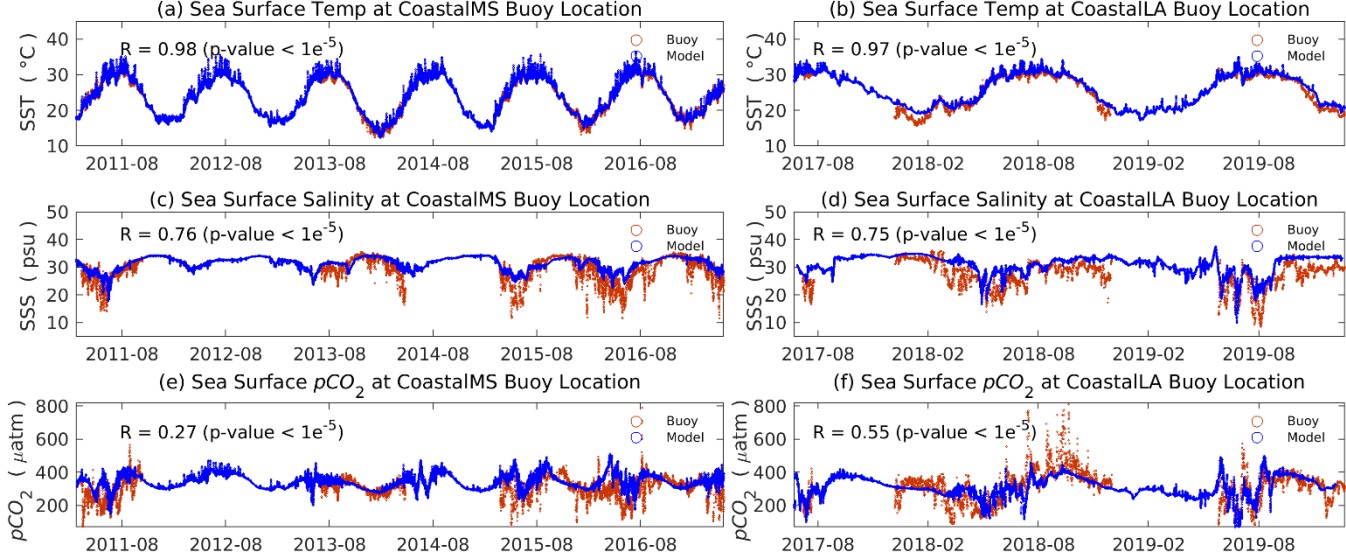

Figure 4: Timeseries of SST, SSS, and $pCO_2$_sea at site CoastalLA and CoastalMS.

## 3.2 Model–cruise comparisons

Cruise carbon measurements include underway water $pCO_2$ data and conductivity–temperature–depth (CTD) bottle results. We compare the model result at the LA transect with the observations of GOMECC cruises conducted at the same location

during GOMECC1 in 2007, GOMECC2 in 2012, and GOMECC3 in 2017, respectively. Measurements of TA during GOMECC cruises followed Dickson's definition (1981), where the TA is expressed as Eq. (3)

$$TA = [HCO_3^-] + 2[CO_3^{2-}] + [B(OH)_4^-] + [OH^-] + [HPO_4^{2-}] + 2[PO_4^{3-}] +$$
$$[H_3SiO_4^-] + [NH_3] + [HS^-] - [H^+] - [HSO_4^-] - [HF] - [H_3PO_4] - [HNO_2] \quad (3)$$

Equation (3) contains fourteen variables, among which $[PO_4^{3-}]$ are explicitly modeled as active tracers, $[HCO_3^-], [CO_3^{2-}], [B(OH)_4^-], [OH^-], [HPO_4^{2-}], [H_3SiO_4^-], [H^+], [HF], [H_3PO_4], [HSO_4^-]$ are calculated by the carbon module, and $[HS^-], [HNO_2], [NH_3]$ are unaccounted. Figure 5 shows the vertical profiles of observed DIC, TA, and their ratio collected at the LA transects (-90°W, 27.5°N - 29.1°N, shown in Fig.1) overlaid with the model solution, the top 200-meter depth is stretched three times to have a better view on the more densely sampled observational data, and a black dot is placed in the location of each observational data with the oversized colored dot representing the value of the measurement. All three transects were taken during July when nutrient supply from the MARS was high. Model simulated profiles at the transects are taken from the closest date of the daily-averaged output. The general trends in Fig. 5 for DIC, TA, and their ratio, demonstrate a good match between the model result and the in situ CTD data. Relative low surface DIC concentration (< 2150 mmol m$^{-3}$) above 200 m isobath demonstrates the river's influence at the NGoM. The general increasing trend of DIC with depth confirms the presence of a biological pump, where inorganic carbon is utilized during photosynthesis in the euphotic layer. Subsequently, the generated OM sinks into deeper waters while being remineralized along the way. The TA profiles show more variation compared with DIC, where generally, a lower TA concentration (< 2380 milliequivalents m$^{-3}$) could be found at the surface as the direct dilution from river discharge, followed by a quick increase to ~2380 milliequivalents m$^{-3}$ in the euphotic layer due to the active photosynthetic activities, which generate alkalinity. Further deep, the TA profiles show a decreasing trend between 200 and 700 m, which could be explained by the water column respiration and nitrification. The TA profiles show a slow increase from 800 m and deeper, which coincides with the alkalinity generation processes in sediment and possibly dissolution of carbonate minerals, both adding to the bottom water TA. The TA/DIC ratio has a maximum at the surface due to low DIC concentration and decreases with depth as DIC concentration increases. The last column in Fig. 5, namely (d) (h) (l), quantifies the difference between the model solutions and the observations and the distribution of the difference. Fig. 5 (d) shows that 51.2% of Model-Obs difference for DIC is within [-10 10] μmol kg$^{-1}$. Similarly, Fig. 5(h) shows that 49.8 % of Model-Obs difference for TA is within [-10 10] μmol kg$^{-1,}$ and Fig. 5(l) shows that 91.6% of Model-Obs difference for TA/DIC ratio is within [-0.02 0.02] unit. The model's RMSE for DIC, TA, and TA/DIC ratio over GOMECC(2007~2017) LA transects dataset is 30.97, 26.86, and 0.014, respectively.

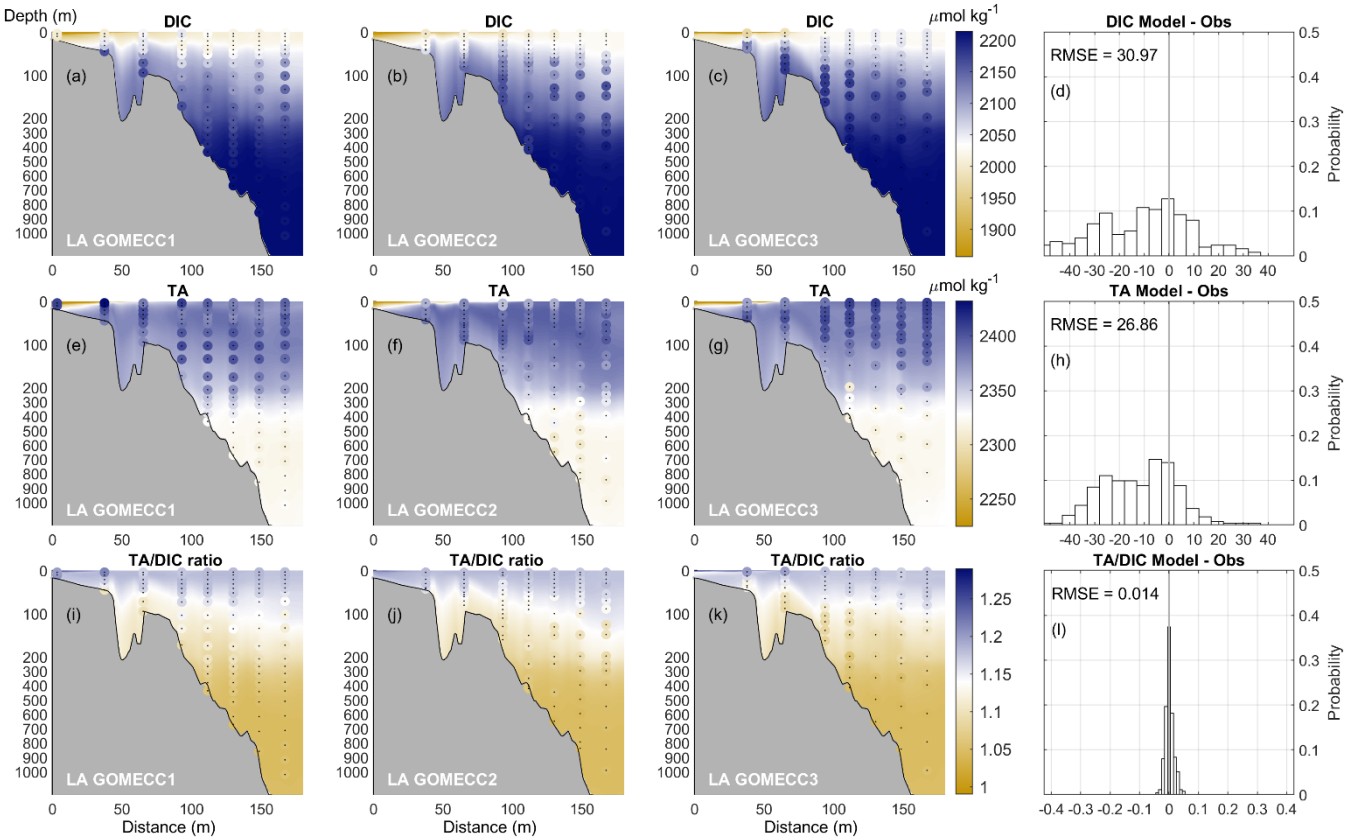

**Figure 5: Discrete measurement of DIC and TA along the LA transect during GOMECC1, GOMECC2, and GOMECC3 cruises shown as oversized scattered dots (with the little black dots indicating their locations), compared with model results in color contour, with the water depth shown on the left side of each figure in meter. The distributions of model bias and RMSE for DIC, TA and TA/DIC ratio combining three GOMECC cruises at LA transect are shown in (d)(h)(l), respectively.**

## 3.3 Model–ML $pCO_2$ product comparisons

Direct comparison between cruise measurement of ocean surface $pCO_2$ and daily averaged model result might suffer from systematic bias due to the sparsity of curies data, both temporally and spatially. The ML model generates surface $pCO_2$ from Chen et al. (2019) integrated >220 cruise surveys between 2002-2019 and MODIS ocean color product covering 2002-2017. The comparison between the two surface $pCO_2$ products is shown in Fig. 6, where surface $pCO_2$ results from Chen et al. (2019) are denoted as "ML" and results from this work are denoted as "Model" for the monthly climatology from July 2002 to December 2017. The two results exhibit a similar spatial distribution of surface $pCO_2$, with our model result revealing more dominant features from the Loop Current in the open ocean. Compared with the ML model, our model produces lower $pCO_2$ estimates over NGoM during winter and fall, higher $pCO_2$ estimates over WF during summer, and stronger influence from the Caribbean Sea. Chen et al. (2019) reported that no satellite data was found for $pCO_2$ <145 μatm or >550 μatm during their

model development. This can also be a factor when considering the differences between the two products. Further comparison between our model and other products can be found in Section 5.1.

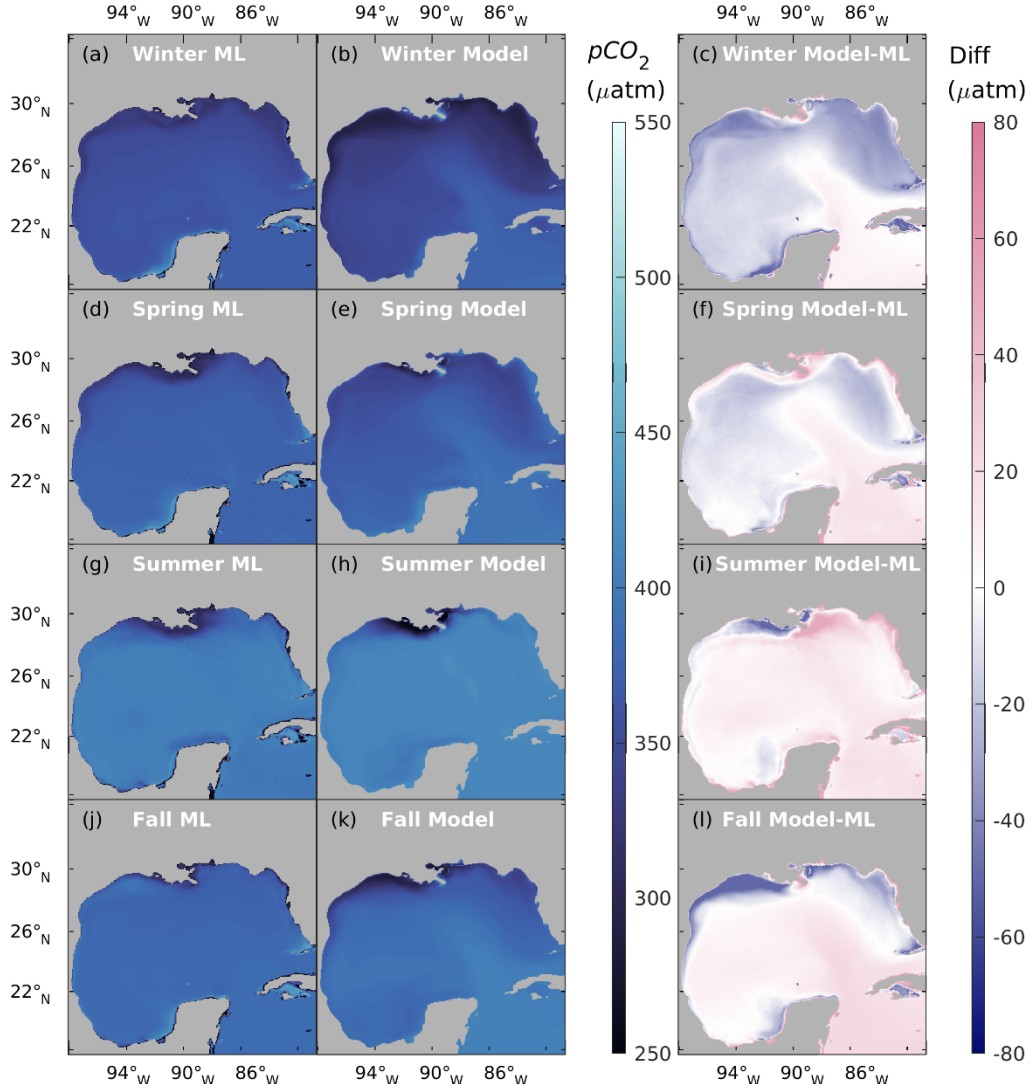

**Figure 6: Comparison of surface *pCO₂* between ML model (Chen et al, 2019) and this work.**

Besides buoy records, transects, and the ML products, we also perform an extensive model-data comparison using available ship-based underway $pCO_2$ measurements from the Ocean Carbon and Acidification Dataset (https://www.ncei.noaa.gov/access/oads/). These extensive model-data comparisons give us the confidence that our model, driven by carbon boundary conditions from the global model, can reproduce temporal, spatial, and vertical variability of the $CO_2$ dynamics in the GoM.

## 4 Result

In this section, we present the spatial and temporal pattern of key carbon system variables, namely $pCO_2$, $pH$, $\Omega$, and air-sea
$CO_2$ flux simulated over the past 20 years in the GoM. In this study, we emphasize the surface carbon condition in two regions:
NGoM and Open GoM, where most existing *in situ* data are distributed. We perform a linear fit of the time series of the key
carbon system variables in each region and show the fitted relationships in Fig. 7. The slopes of the fitted linear plot give
estimations of the change rate of each carbon variable over the past two decades.

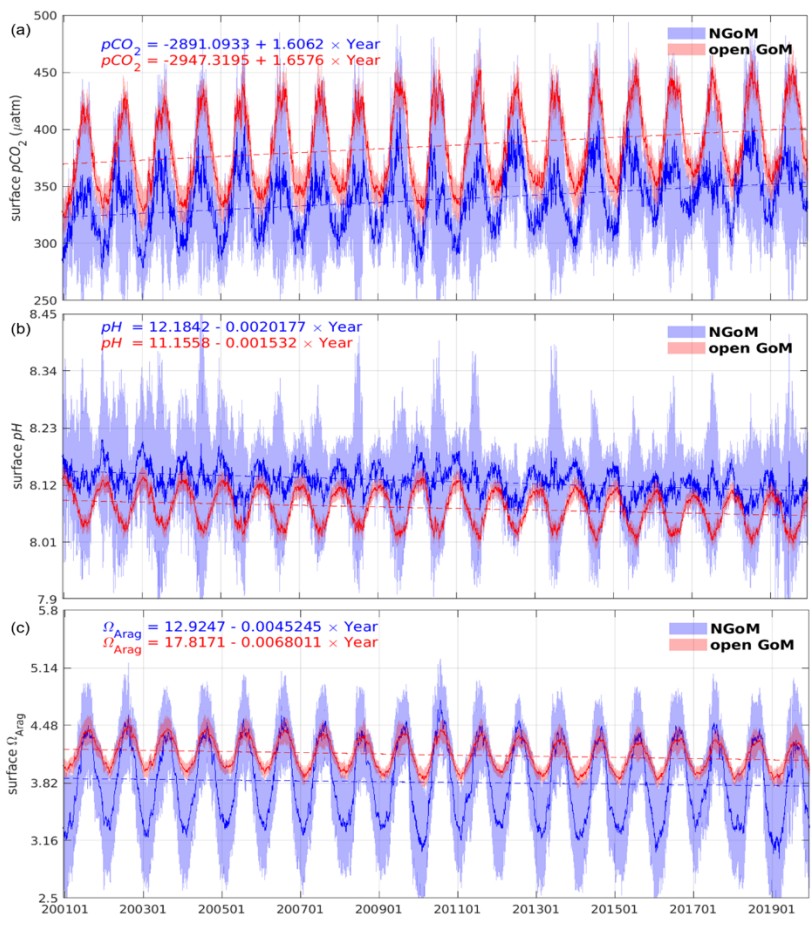

Figure 7: Time series and trend analysis of sea surface (a) $pCO_2$, (b) $pH$, (c) $\Omega_{Arag}$ for the NGoM (blue), Open GoM (red). Solid
lines depict the daily spatial mean value; shaded areas stand for one standard deviation, and dash lines trace the linear fit of the
time series.

### 4.1 $pCO_2$

We simulate a generally increasing trend in surface $pCO_2$ level for both NGoM and Open GoM, with an increasing rate of 1.61
$\mu$atm yr$^{-1}$ and 1.66 $\mu$atm yr$^{-1}$, respectively (Fig. 7). Seasonal ocean surface $pCO_2$ variation is primarily affected by temperature
variations. To evaluate the $pCO_2$ trend without temperature effects, we decouple the thermal and nonthermal components of

$pCO_2$ at the ocean surface using Eq. (4) and (5) and further extract the $pCO_2$ variation due to gross primary production and air-sea $CO_2$ flux. The temperature sensitivity of $CO_2$, $\gamma_T = 4.23\%$ per degree Celsius, proposed by Takahashi et al. (1993), is used to perform the thermal decoupling in Eq. (4) and Eq. (5). The thermal effect on $pCO_2$ ($pCO_2^{th}$) is defined as the deviation between apparent $pCO_2$ and the estimated $pCO_2$ at the mean SST (denoted as <SST>). The nonthermal counterpart ($pCO_2^{nt}$) is obtained by removing the thermal effect from the $pCO_2$ time series using Eq. (5). Note that this definition of $pCO_2^{th}$ is different from the original definition given by Takajashi et al. (2002). The new definition allows the thermal and nonthermal $CO_2$ components to sum up to the apparent $pCO_2$. $pCO_2$ variations due to gross primary production are estimated from the carbon module based on the DIC consumed by gross primary production and denoted as $pCO_2^{GPP}$. $pCO_2$ variations due to air-sea $CO_2$ flux are calculated from the carbon module based on the DIC change from the air-sea exchange and denoted as $pCO_2^{flux}$.

The contribution from gross primary production (GPP) is the process that directly affects the $CO_2$ uptake, and GPP can be calculated by tracking the photosynthesis activity of diatom and small phytoplankton (which is a function of solar radiation, temperature, nutrients, and phytoplankton concentrations). Respiration, on the other hand, is more complicated to quantify since it concerns both living biota (phytoplanktons, zooplanktons) and nonliving detritus (PON, DOC). Both respiration at the surface and respiration that happens in deeper water as detritus sink modify DIC concentration and create concentration gradients. We leave the respiration in the end-member of the $pCO_2^{nt}$ components, which incorporated various mixing processes (e.g. river water and oceanic water mixing, vertical mixing of upwelled waters, horizontal advection induced lateral transport of tracers with concentration gradients, and entrainment of waters with different chemical nature (i.e. temp/ salt/ DIC/ TA/ detritus concentration)). Remineralization and respiration are included in the term $pCO_2^{mixing}$ due to the result of the two processes altering water chemical nature (DIC, TA, detritus concentration) and the impacts of water chemical nature on $pCO_2$ are constantly being modified by (and as a result of) the mixing process.

$$pCO_2^{th} = pCO_2 \cdot [1 - exp(\gamma_T \cdot (< SST > - SST))] \quad (4)$$

$$pCO_2^{nt} = pCO_2 \cdot exp(\gamma_T \cdot (< SST > - SST)) \quad (5)$$

$$pCO_2^{nt} = pCO_2^{GPP} + pCO_2^{flux} + pCO_2^{mixing} \quad (6)$$

Figure 8 shows the seasonal and spatial patterns of four decoupled $pCO_2$ components, namely the $pCO_2^{th}$, $pCO_2^{nt}$, $pCO_2^{GPP}$, and $pCO_2^{flux}$. The $pCO_2^{th}$ patterns in the second row (e,f,g,h) of Fig. 8 reflect the fluctuation of $pCO_2$ due to thermal effects. Over the four seasons, a general pattern of rising $pCO_2^{th}$ from spring to summer and a gradual reduction from summer onwards can be observed. The NGoM shelf exhibits the lowest $pCO_2^{th}$ values during winter, while WF shows elevated $pCO_2^{th}$ values during summer. The higher $pCO_2^{th}$ values dwelling in the southern part of the Yucatan shelf reveal the warm water flowing into the GoM from the Caribbean Sea. The top row (a,b,c,d) of Fig. 8 shows the nonthermal component of $pCO_2$. The relatively high $pCO_2^{nt}$ during winter on the NGoM shelf, compared to that of the Open GoM, shows the strong solvation effect of $CO_2$ with low SST, contributing to a high DIC/TA ratio and strong carbon uptake. The lower two rows of

Fig. 8 show $pCO_2$ changes

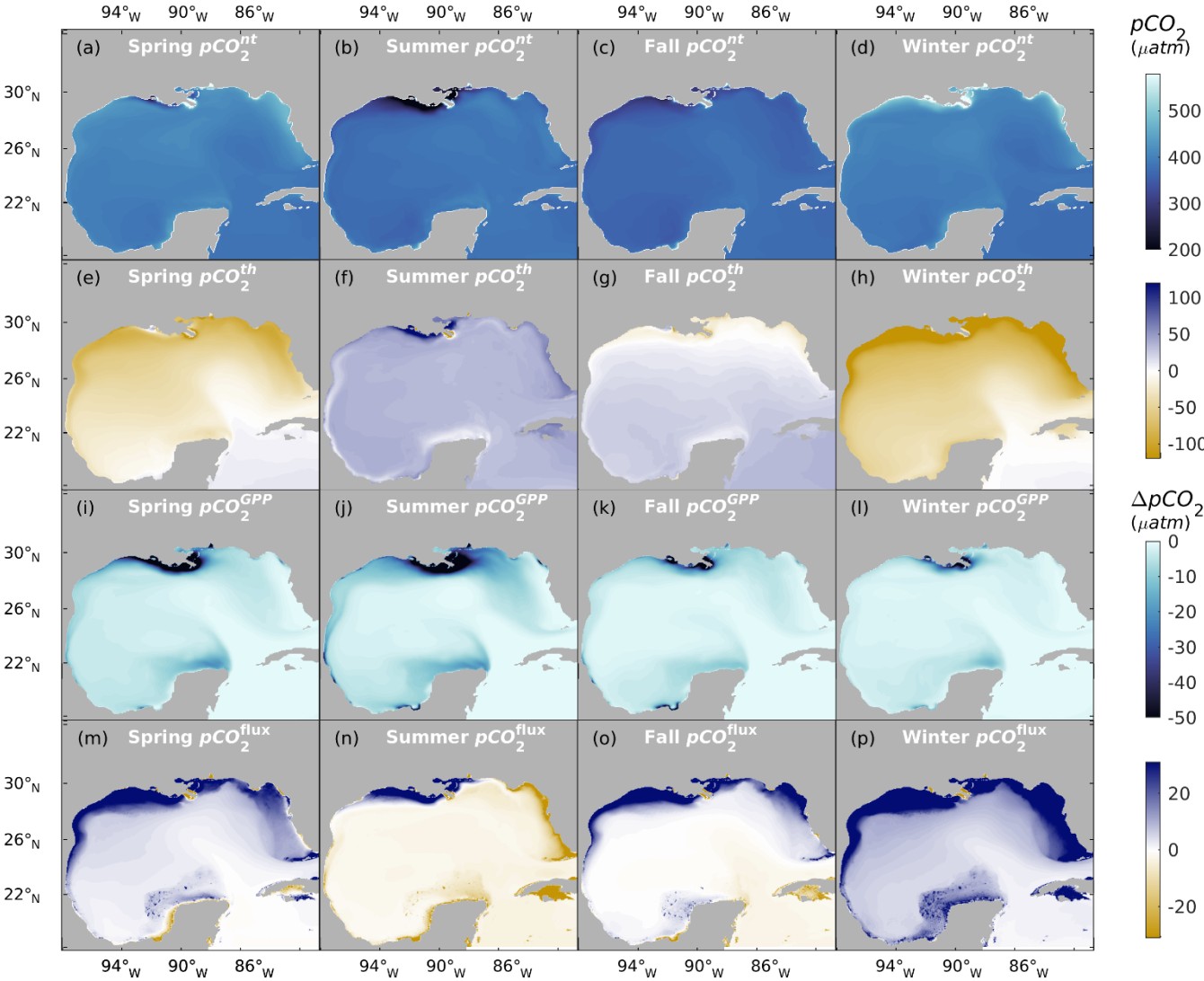

**Figure 8: Spatial distribution of sea surface $pCO_2$ over four seasons. From the top to the bottom row: $pCO_2{}^{th}$ (a through d), $pCO_2{}^{nt}$ (e through h), $pCO_2{}^{GPP}$ (i through l), $pCO_2{}^{flux}$ (m through p; positive indicates the air-sea $CO_2$ flux works in the direction of increasing sea surface $pCO_2$).**

due to the gross primary production and $CO_2$ air-sea exchange, respectively. The $pCO_2{}^{GPP}$ reflects the intensity of primary production in terms of $pCO_2$ reduction. $pCO_2{}^{GPP}$ has larger magnitudes in NGoM during spring and summer and is gradually attenuated during and after fall. The large magnitudes of $pCO_2{}^{GPP}$ during summer in NGoM waters and the Open GoM region following the extension of the MARS plume suggest strong biological $CO_2$ removal in those regions. These results show that gross primary production has a stronger regulation on surface $pCO_2$ during spring and summer in river-dominated waters and

upwelling regions. At the same time, a minor contribution from gross primary production can be seen during winter, on the flat and shallow WF, and in the Open GoM regions southern of Loop Current. The $pCO_2^{flux}$ reflects the intensity of air-sea

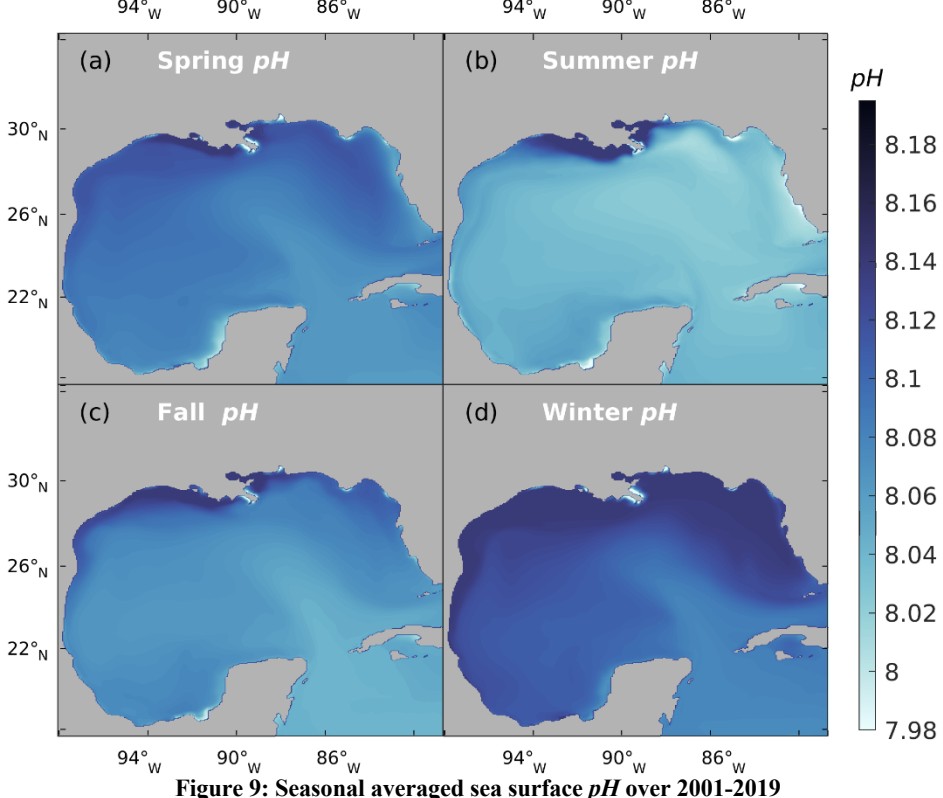

**Figure 9: Seasonal averaged sea surface *pH* over 2001-2019**

CO$_2$ exchange attempting to mitigate the disequilibrium caused by local physical and biological processes. The relatively high value of $pCO_2^{nt}$ and low magnitude of $pCO_2^{GPP}$, as well as low river discharge (minimal river water mixing) in the WF during

winter, indicate a strong CO$_2$ uptake from the atmosphere due to low SST. This analysis agrees with the low $pCO_2^{th}$ and the high $pCO_2^{flux}$ values in the WF during winter, shown in Fig. 8(d,p). Situations during seasons other than winter are more complicated due to active biological activities and mixing. The Mississippi Delta region has a high $pCO_2^{th}$ value during summer, however, combined with the effects of mixing and strong primary production (large magnitude of $pCO_2^{GPP}$), this region acts as a strong carbon sink that exhibits a high value of $pCO_2^{flux}$ compensated from the atmosphere. Figure 8

demonstrates that most of the time around the year, the surface $pCO_2$ pattern is not dominated by a single factor but a combination of multiple controlling processes. The result of $pCO_2$ decomposition agrees with the current view of the $pCO_2$ dynamic and carbon uptake patterns in the GoM, which is strong carbon uptake during winter due to thermal effect and high biological CO$_2$ drawdown during spring and summer under the riverine influence.

### 4.2 *pH*

Ocean surface *pH* in the GoM shows a clear decreasing trend, with a 0.0020 yr$^{-1}$ decrease over the NGoM region and a 0.0015 yr$^{-1}$ decrease over the Open GoM region. Figure 9 shows the seasonal pattern of ocean surface *pH* over the GoM. Spatial and seasonal *pH* patterns show larger variation over the NGoM, especially on the inner shelf (depth<50m). The *pH* level in the surface water is closely associated with temperature, photosynthetic activities, and water mixing. The high *pH* value on the NGoM shelf reveals the strong influence of riverine alkalinity export and nutrient-stimulated primary production. The lower

*pH* values on the WF shelf during summer and the generalized greater *pH* values over NGoM during winter demonstrate the high *pH* sensitivity on SST. The upwelling region along the west Yucatan shelf shows reduced *pH* values all year round compared with its surrounding waters. The upwelling along the WGoM slope has a similar effect of reducing and maintaining a relatively low *pH*, effectively forming a *pH* boundary between the shelf water and the Open GoM. The Open GoM is largely dominated by the warmer and lower *pH* water from the Caribbean Sea throughout the year.

### 4.3 Aragonite and Calcite Saturation State

Aragonite undersaturation occurs ($[CO_3^{2-}] < 66$ μmol kg$^{-1}$) before calcite undersaturation ($[CO_3^{2-}] < 42$ μmol kg$^{-1}$) (Feely, Doney, & Cooley, 2009; Feely et al., 2002). As a result, $\Omega_{Calc}$ is approximately 50% higher than $\Omega_{Arag}$, and their spatial and seasonal variations are very similar, as shown in Fig. 10. Variations in temperature, alkalinity, and $pCO_2$ impose important controls on $\Omega_{Arag}$. The multiyear variability of $\Omega_{Arag}$ at the ocean surface is shown in Fig. 7(c). The NGoM region shows a

smaller decreasing trend in $\Omega_{Arag}$ (0.0045 yr$^{-1}$) compared to that of Open GoM (0.0068 yr$^{-1}$). Noted the data contained in Fig. 7 does not include water from the shallow shelf waters (water depth < 10 m), therefore, the trend in NGoM does not incorporate the condition in coastal estuaries. The spatial distribution of $\Omega_{Arag}$ across the GoM depicts a healthy level of $\Omega$ and a low risk of ocean acidification (Fig. 10). While the coastal ocean generally has a relatively high $\Omega$ level, some coastal locations warrant special attention when evaluating their tendency towards calcium mineral dissolution. These locations include coastal regions

that experience a large load of riverine OM inputs (e.g., the Mississippi River delta in summer) and the upwelling regions that receive relatively higher acidity water from the bottom ocean (e.g., west of Yucatan). These regions show significant $\Omega$ reductions compared to the surrounding waters and are potential victims of ocean acidification. The influence of the river on $\Omega$ is complex. On one hand, a high nutrient level of river discharge could stimulate a high photosynthetic rate which consumes DIC and increases $\Omega$. On the other hand, photosynthesis favors calcification which consumes carbonate ions and reduces $\Omega$.

Therefore, $\Omega$ is subject to increase with stronger photosynthesis and decrease with stronger calcification. Hence the magnitude of the overall effect will depend upon photosynthetic rates and the calcification rate. In this work, two phytoplankton groups are modeled, diatom (silicifying) and small phytoplankton (implicitly including the calcifying coccolithophores, foraminifera, and dinoflagellates), of which only the small phytoplankton group has the potential to conduct calcification (Raven & Giordano, 2009). Besides being regulated by temperature and small phytoplankton concentration, the calcification rate also

depends on the composition of the phytoplankton population. The small phytoplankton group has a survival advantage at

relatively low nitrogen concentrations and could be grazed by two zooplankton groups (meso-zooplankton and micro-zooplankton), whereas diatom is more nutrient-demanding and can be grazed by three zooplankton groups (predator zooplankton, meso-zooplankton, and micro-zooplankton). The competitive phytoplankton evolution shaped the relative rates between photosynthesis and calcification on the NGoM shelf during summer. The reduced $\Omega$ to the east of the Mississippi River delta is a combined result of high DIC water entrained by Loop Current eddies west of the delta and an increased ratio of small phytoplankton in offshore waters.

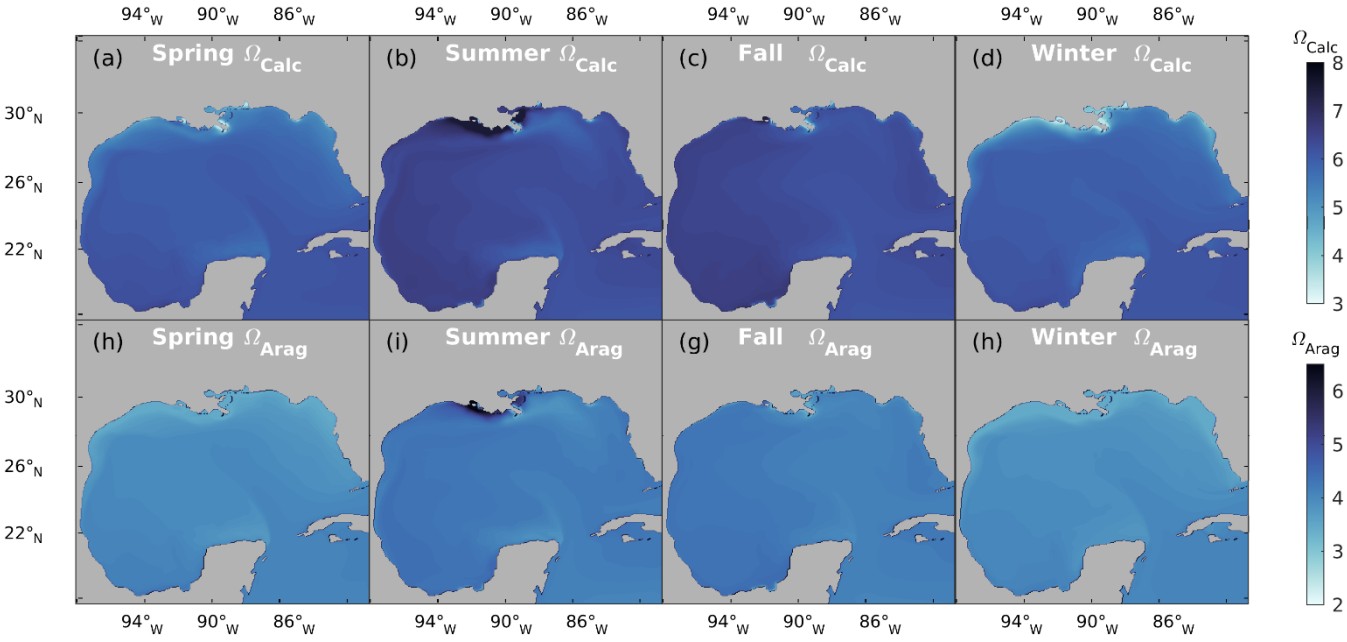

**Figure 10: Seasonal mean sea surface $\Omega_{Arag}$ and $\Omega_{Calc}$ over 2001-2019**

## 4.4 Air-sea CO₂ Flux

Air-sea $CO_2$ flux is calculated from daily averaged model data from 2001 to 2019 (Table 3). The GoM overall is a $CO_2$ sink with a mean flux rate of 0.62 mol C m⁻² yr⁻¹ (11.77 Tg C yr⁻¹), which is commensurate with the reported value of 11.8 Tg C yr⁻¹ by Coble et al. (2010). The greatest carbon uptake rate occurs in winter (1.97 mol C m⁻² yr⁻¹), while the weakest carbon uptake is present in fall (0.16 mol C m⁻² yr⁻¹). The strongest carbon efflux is simulated in summer (-0.57 C m⁻² yr⁻¹). On average, water in the NGoM acts as a sink throughout the years, and the water in the Open GoM acts as a weak source during summer (and fall for 2002, 2004, 2006, and 2009) and a sink during the rest of the year. The direction and magnitude of the air-sea exchange can be seen in Fig. 11, where a positive number indicates the ocean is a carbon sink. The NGoM is a very strong $CO_2$ sink year-round, and Open GoM is a source of $CO_2$ during summer but a sink in the rest of the year (except during fall in a few years), as shown in Fig. 11(b). There are clear trends and patterns in multiyear $CO_2$ air-sea flux, as shown in Fig. 11(a), where a greater air-sea $CO_2$ flux average could be seen at the end of 2019 than that of 2001, resulting in a stronger

carbon influx in both regions. A significant anomaly in the middle part of the record (2009-2011) can be observed, which could result from the influence of a large negative North Atlantic Oscillation (NAO) indice and El Niño in 2010 (Buchan et al., 2014). Similar observations in the Caribbean Sea are attributed to the single-year anomalies in the climate indices and the climate mode teleconnection (Wanninkhof et al., 2019).

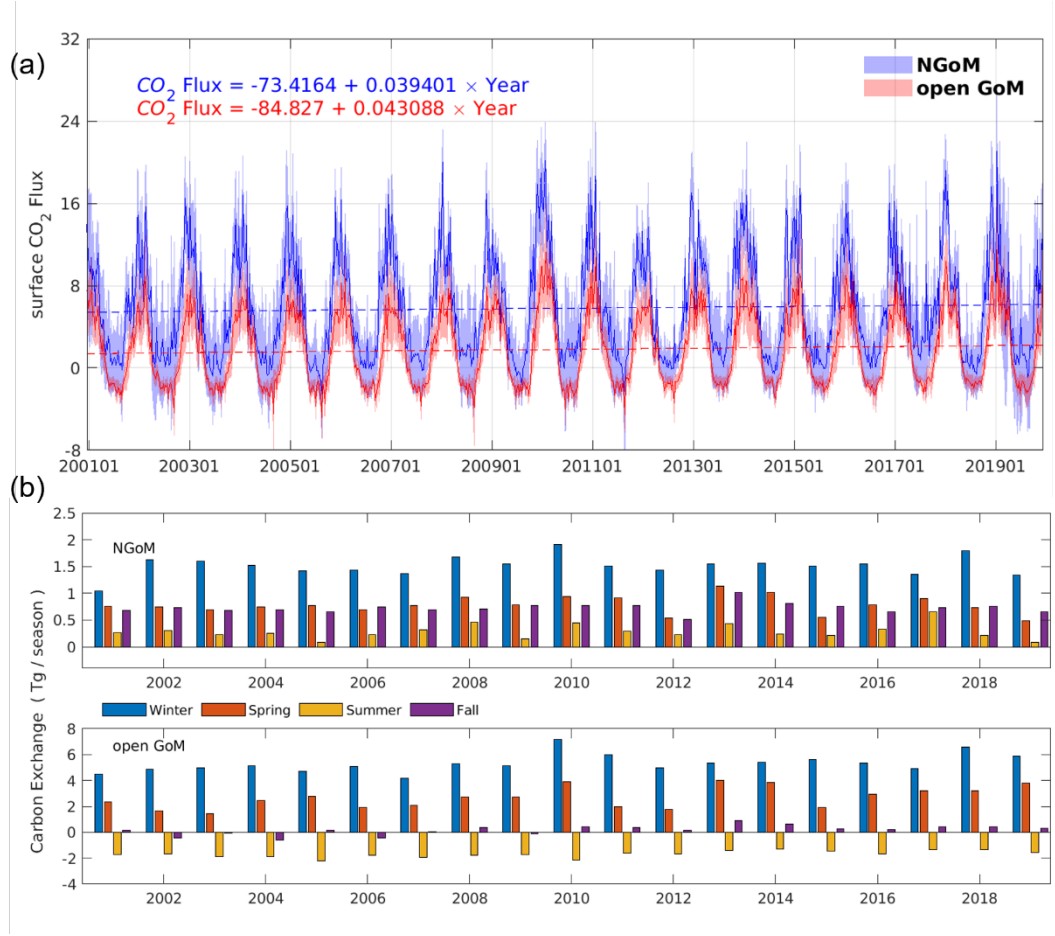


**Figure 11: Air-sea CO₂ exchange over GoM: a) multiyear CO₂ flux regression over the weekly mean levels in NGoM and Open GoM; b) seasonal CO₂ air-sea exchange budget over two decades in NGoM and Open GoM.**

### 4.5 Contribution from River and Global Ocean

In this session, we further diagnose river discharge and the global ocean's impacts on the GoM carbon system via a comparison between the control experiment (His) and the two perturbed experiments (Bry and NoR). In the Bry expreiment the clamped boundary conditions that repeat the DIC and TA level as that of the year 2000, and in the NoR experiment the river forcing was removed to examine the impact of fluvial input on the coastal carbon system.

Figure 12 shows the multiyear mean levels of the four carbon variables ($pCO_2$, $pH$, $\Omega_{Arag}$, and $CO_2$ flux) simulated by the three experiments. Table 1 summarizes the mean levels of $pCO_2$ over the NGoM and Open GoM. The definition of the $pCO_2{}^{th}$, the $pCO_2{}^{nt}$, the $pCO_2{}^{GPP}$, the $pCO_2{}^{flux}$, and the $pCO_2{}^{mixing}$ can be found in Sect. 4.1. The $pCO_2{}^{mixing}$ is defined in Eq. (6), which reflects the $pCO_2$ level due to the water mixing. It can also be considered as the $pCO_2$ level determined by the water with a multiyear mean temperature and without the influence of gross primary production or air-sea $CO_2$ flux.

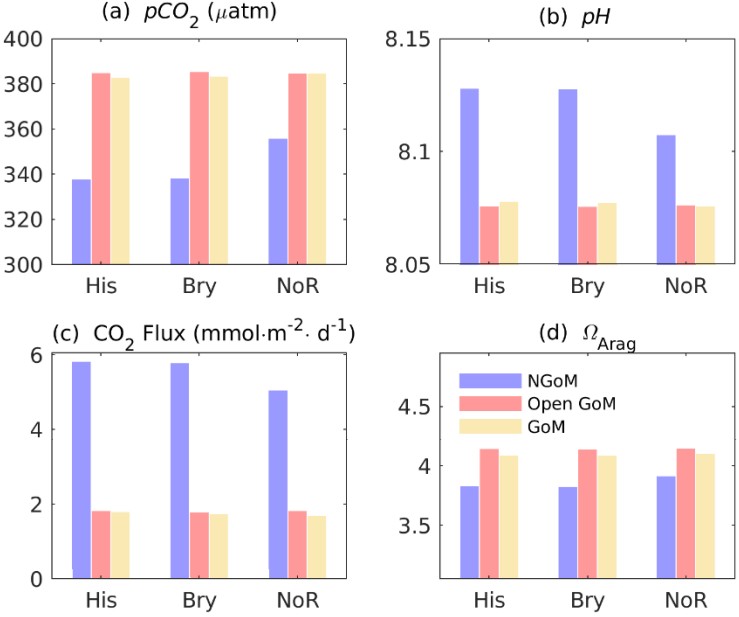


**Figure 12: Multiyear synoptic of sea surface $pCO_2$, $pH$, $CO_2$ air-sea flux, $\Omega_{Arag}$ with His, Bry, and NoR experiment. Color bars show the corresponding mean level from 2001 to 2019. Color legend: blue – NGoM, red – Open GoM, yellow – GoM wide. The unit for $pCO_2$, $pH$, $CO_2$ flux, and $\Omega_{Arag}$ are µatm, 1 (full scale), mmol·m$^{-2}$·d$^{-1}$, 1, respectively. (Noted that the $CO_2$ air-sea flux used ocean convention, where a positive value indicates transport from air to sea, i.e., the ocean is a sink)**

The most salient difference among the three experiments is the significant elevation of the annual mean $pCO_2$ level (in Fig. 12) in the NGoM by the NoR experiment, combined with a significantly reduced carbon sink during summer (in Fig. 13, from 0.287 to -0.093 Tg season$^{-1}$ using His as a benchmark). The difference can be better resolved by the $pCO_2$ decomposition results shown in Table 1, where a drastic change in the water carbon system emerges in the NGoM during the NoR experiment (as compared to the other experiments with river input), evidenced by the large $pCO_2{}^{mixing}$ value deviated from that of His and

Bry experiment in the NGoM region during spring, summer, and winter. The low values of $pCO_2{}^{nt}$ in NGoM during summer can be explained by a strong biological drawdown of $CO_2$ associated with the high productivity fuelled by the riverine nutrient supply. A $pCO_2{}^{GPP}$ component of -35.35 and -35.46 µatm corresponding to the strong biological drawdown of $CO_2$ in Bry and His experiments are in sharp contrast to that of the NoR experiment, which is only -3.10 µatm. Consequently, distinct patterns of $CO_2$ air-sea flux are shown in Fig. 13, and highly contrasting $CO_2$ air-sea flux-induced surface $pCO_2$ changes are shown in

Table 1 ($pCO_2{}^{flux}$). The summer $pCO_2{}^{flux}$ component for NGoM of the two experiments with river inputs exhibits a relatively large value (~ 43 µatm) compared with that of the NoR experiment (0.2 µatm), demonstrating a much smaller disequilibrium

between oceanic and atmospheric $pCO_2$ when rivers are absent. The changes introduced by removing the river showcase the dominating impact of river input on the NGoM carbon system in terms of gross primary production, surface $pCO_2$ level, and air-sea $CO_2$ exchange. Due to the different intensities of gross primary production, in His experiment sediment PON concentration is six times that of the NoR experiment, and riverine nutrients in His fostered a ~105 times higher PON burial rate in NGoM sediments than that of NoR.

Table 1 show a close resemblance in the magnitude and seasonal pattern between Bry and His experiment in the Open GoM region, with a small yet steady reduction in $pCO_2^{\text{mixing}}$ by the Bry experiment among all seasons. The small reductions in $pCO_2^{\text{mixing}}$ of the Bry experiment compared to that of His reflect the contribution from extraneous carbon accumulation from the global ocean that is included in the His experiment. As expected, slightly greater $CO_2$ sink values are reported in Fig. 13 for Bry than His. Since the Bry experiment has a smaller carbon accumulation in the Open GoM region compared to that of the His experiment, the ocean surface requires a slightly greater carbon uptake to reach equilibrium with the atmosphere. Since oceanic water is a natural buffer system and the ocean surface is under constant interaction with the atmosphere, it is reasonable that the His and Bry experiments do not show significant differences in surface carbon variables. However, this does not mean the accumulative signal of DIC from the global ocean is neglectable. As shown in Fig. 14, the ocean water equilibrium witnessed migration over the 20-year simulation. Compared with the Bry experiment, His experiment received accumulated carbon input from the global ocean and underwent a $[CO_3^{2-}]$ reduction as large as 15% in some inflicted regions at the 100 m depth. Combining the results from the three experiments, we conclude that, in addition to elevated atmospheric $CO_2$ levels, both inputs from MARS and global oceans contribute to the overall acidification trend in GoM, with the impacts from MARS mainly limited to the NGoM shelf region and the global ocean's impacts spanning in the Open GoM.

**Table 1.** Sea surface $pCO_2$ decomposition among experiments

| | unit: $\mu atm$ | NGoM | | | Open GoM | | |
|---|---|---|---|---|---|---|---|
| | | His | Bry | NoR | His | Bry | NoR |
| Spring | $pCO_2$ | 356.73 | 356.20 | 351.04 | 371.55 | 371.13 | 371.12 |
| | $pCO_2^{nt}$ | 432.75 | 432.11 | 423.55 | 399.59 | 399.14 | 399.09 |
| | $pCO_2^{th}$ | -76.03 | -75.92 | -72.51 | -28.04 | -28.01 | -27.98 |
| | $pCO_2^{GPP}$ | -26.83 | -26.79 | -3.34 | -2.95 | -2.96 | -2.64 |
| | $pCO_2^{flux}$ | 56.83 | 57.41 | 49.24 | 3.41 | 3.46 | 3.46 |
| | $pCO_2^{mixing}$ | 401.72 | 400.43 | 376.06 | 399.12 | 398.64 | 398.27 |
| Summer | $pCO_2$ | 369.53 | 368.98 | 413.42 | 417.45 | 417.00 | 415.42 |
| | $pCO_2^{nt}$ | 327.13 | 326.63 | 364.24 | 382.49 | 382.09 | 380.7 |
| | $pCO_2^{th}$ | 42.41 | 42.35 | 49.18 | 34.95 | 34.91 | 34.72 |

| | | | | | | |
|---|---|---|---|---|---|---|
| | $pCO_2^{GPP}$ | -35.46 | -35.35 | -3.10 | -3.44 | -3.36 | -2.09 |
| | $pCO_2^{flux}$ | 43.41 | 43.57 | 0.20 | -2.38 | -2.34 | -2.19 |
| | $pCO_2^{mixing}$ | 318.90 | 318.13 | 367.24 | 388.31 | 387.79 | 384.97 |
| Fall | $pCO_2$ | 345.82 | 345.56 | 352.78 | 393.38 | 393.12 | 392.80 |
| | $pCO_2^{nt}$ | 353.69 | 353.43 | 358.06 | 370.78 | 370.54 | 370.28 |
| | $pCO_2^{th}$ | -7.8786 | -7.88 | -5.28 | 22.60 | 22.58 | 22.52 |
| | $pCO_2^{GPP}$ | -13.64 | -13.61 | -1.72 | -1.69 | -1.62 | -1.27 |
| | $pCO_2^{flux}$ | 63.87 | 64.08 | 58.23 | 0.19 | 0.23 | 0.25 |
| | $pCO_2^{mixing}$ | 301.73 | 301.22 | 299.60 | 372.28 | 371.94 | 371.29 |
| Winter | $pCO_2$ | 322.56 | 322.32 | 305.96 | 348.71 | 348.20 | 348.80 |
| | $pCO_2^{nt}$ | 458.11 | 457.77 | 432.23 | 393.36 | 392.80 | 393.45 |
| | $pCO_2^{th}$ | -135.55 | -135.45 | -126.27 | -44.65 | -44.59 | -44.65 |
| | $pCO_2^{GPP}$ | -8.01 | -8.06 | -1.88 | -1.77 | -1.82 | -1.67 |
| | $pCO_2^{flux}$ | 89.62 | 90.03 | 121.45 | 6.85 | 6.92 | 6.83 |
| | $pCO_2^{mixing}$ | 373.68 | 372.96 | 308.51 | 388.27 | 387.69 | 388.29 |

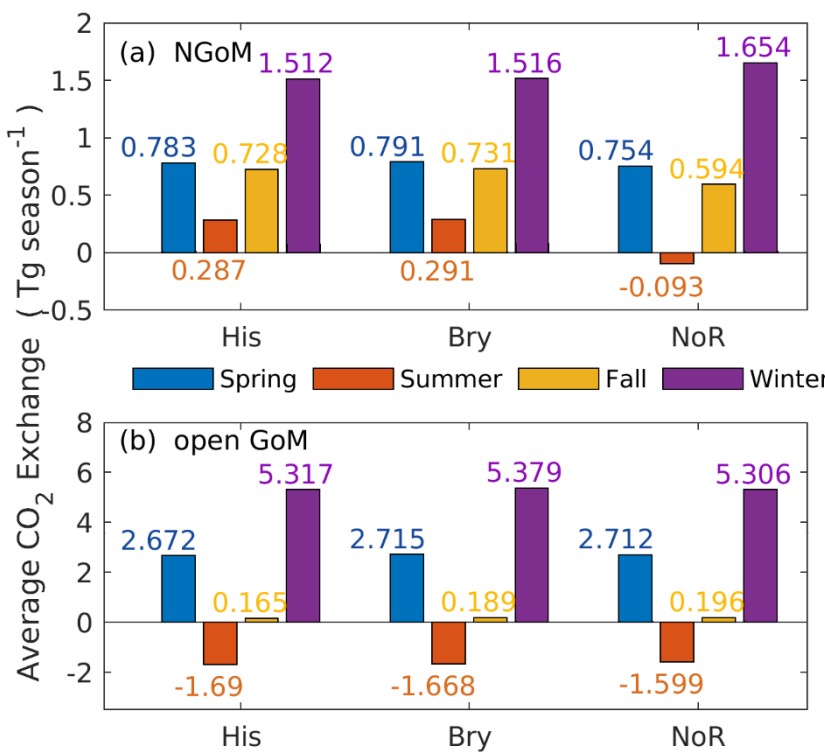

**Figure 13: Seasonal air-sea CO₂ exchange at NGoM and Open GoM region among the His (historical), Bry (fixed boundary), and NoR (non-river) experiments.**

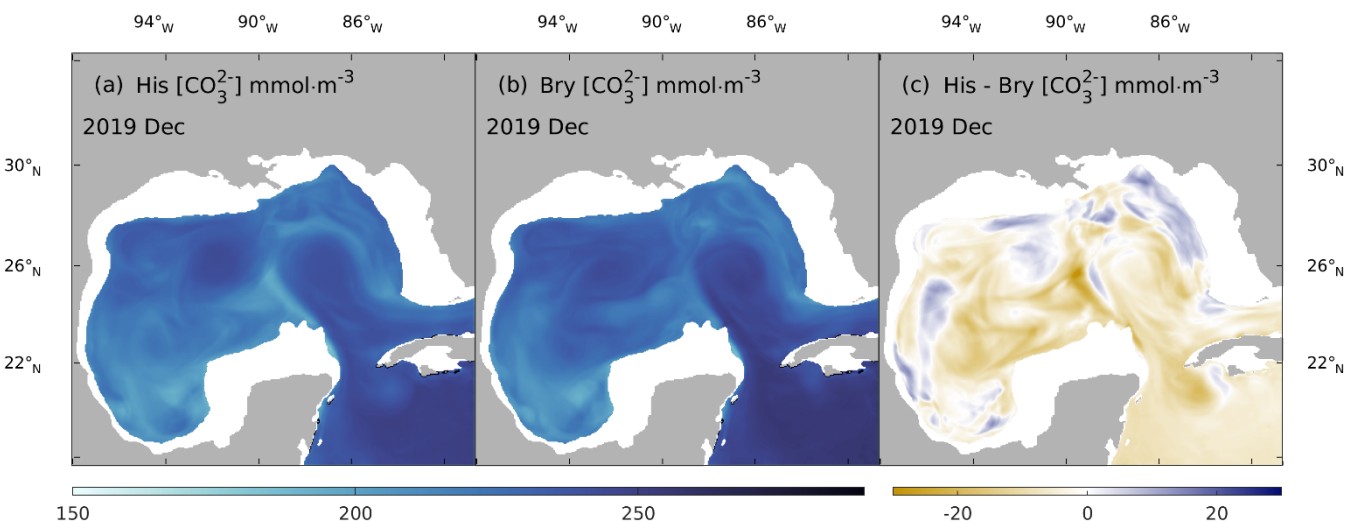

**Figure 14: Comparison of monthly averaged carbonate ion concentration ( [CO₃²⁻] ) between His and Bry at 100 m depth in 2019. (a) December mean [CO₃²⁻] of the His experiment, (b) December mean [CO₃²⁻] of the Bry experiment, and (c) difference between (a) and (b).**

## 5 Discussion

In this study, we demonstrate that the regional high-resolution carbon model can reproduce the spatial and seasonal patterns of ocean surface $pCO_2$ in the GoM and generate reliable TA/DIC profiles in the NGoM shelf. We detect a consistent acidification trend in the GoM over the past two decades. In this section, we present a side-by-side evaluation of the regional model with GCMs, global climatology products, and other regional models, followed by an envisioning of the future outlook and model development.

## 5.1 Model Performance

In this section, we further evaluate the performance of our model via comparison against different global and regional models, and climatological products. The GoM region has limited observations of dissolved inorganic carbon (DIC) and total alkalinity (TA), and observational data covering different depths are even fewer. We examine several climatology products downloaded from the National Centers for Environmental Information (NCEI) data service and include the NCEI Accession number for quick reference of each product. Due to lack of observation, global climatology products either have no coverage in the GoM region, e.g., Mapped Observation-Based Oceanic DIC monthly climatology from the Max-Planck-Institute for Meteorology (MOBO-DIC_MPIM) (NCEI Accession 0221526) (Keppler et al. 2020) or only contain surface carbon variables, e.g., global

gridded data set of the surface ocean carbonate system OceanSODA-ETHZ (v2021,NCEI Accession 0220059) (Gregor and

Gruber, 2020), Climatological Distributions of $pH$, $pCO_2$, Total $CO_2$, Alkalinity, and $CaCO_3$ Saturation in the Global Surface Ocean (NCEI Accession 0164568) (Takahashi et al. 2017), the partial pressure of carbon dioxide collected from Surface underway observations in the world-wide oceans (NCEI Accession 0161129) (Bakker et al. 2017), an observation-based global monthly gridded sea surface $pCO_2$ product (NCEI Accession 0160558) (Landschützer et al., 2017), and a global ocean $pCO_2$ climatology combining open ocean and coastal areas (NCEI Accession 0209633) (Landschützer et al. 2020). The most updated

global monthly TA (NCEI Accession 0222470) (Broullón et al. 2020b) and DIC (NCEI Accession 0222469) (Broullón et al. 2020a) products offer a 12-month climatology with a 1º x1º spatial resolution and 102 vertical levels. Nevertheless, these products utilized a neural network approach to achieve the 3-dimensional coverage. Thus, one should be cautious that the generated monthly climatology products are not built solely from the interpolation of observation. Rather, they are machine learning products with many untested assumptions. For instance, they used $pCO_2$ from LDEOv2016 (Takahashi, Sutherland,

Kozyr, 2017) and TA from Broullón et al. (2019) to compute surface DIC values to increase the spatial coverage in the training data used by the machine learning model (Broullón et al. 2020a,b). In contrast, GCMs are based on large-scale circulations that are coupled with biogeochemical processes. They utilize rigorous reasoning numerical methods with conservation schemes and, therefore, should have higher inherent consistency. In the following, we check the bias of these climatology products and GCMs products (see Fig. 15) using Mean Bias, RMSE and R-value defined as follows:

$$Mean\ Bias = \sum_{i=1}^{N}(M_i - O_i)/N \qquad (7)$$

$$RMSE = \sqrt{\frac{1}{N}\sum_{i=1}^{N}(M_i - O_i)^2} \qquad (8)$$

$$R\text{-}value = \frac{Cov(M,O)}{\sigma_M \sigma_O} \qquad (9)$$

where M stands for model output, and O stands for observation; Cov refers to the coverance, and $\sigma$ indicates the standard deviation. Further, we utilized the Taylor diagram to assess the model's ability to capture spatial patterns with regard to a given

set of reference data (Babaousmail et al., 2021). The Taylor Skil Score ($TSS$) is defined by Eq. (10).

$$TSS = \frac{4(1+R)^2}{\left(\frac{\sigma_o}{\sigma_m}+\frac{\sigma_m}{\sigma_o}\right)^2(1+R_0)^2} \qquad (10)$$

where $\sigma_o$ and $\sigma_m$ are the standard deviation of observation and model, respectively. The value of $TSS$ range from 0 to 1, with values close to 1 corresponding to better performance. $R$ is the correlation coefficient between the observation and model, and

$R_o$ is the maximum correlation coefficient attainable (we use 0.999).

The non-dimensional model *skill*, defined in Eq. (11)., can be used to quantify the improvement of the model to reproduce observed data with regard to the climatological value.

$$skill = 1 - \frac{\sum_{i=1}^{N}(d_i - \Im[m_i])^2}{\sum_{i=1}^{N}(d_i - c_i)^2} \qquad (11)$$

where $d_i$ are the available measurements, and ($d_i$ -ℑ[$m_i$]) is the observation-model difference, ($d_i$ -$c_i$) is the observation-
climatology difference (Zhang et al., 2012). Usually, a skill of 0.25 means the model can reproduce 25% more variance than
those already described in climatology. By using the observation-model difference from this work and the observation-
climatology difference from other models/climatology, we can evaluate the relative performance between this work and other
model/climatology – a positive *skill* value ideally indicate improvement of the model in the numerator over the
model/climatology in the denominator while a negative *skill* value indicates the opposite. We use the other products as the
reference to calculate the observation-product difference in the denominator, use our model to calculate the observation-model
difference in the numerator, and list the corresponding *skill* value in Table 2, indicating the percentage
improvement/deterioration gained by this work over the referenced product.

In Fig. 15, we interpolated model results (both global and regional) to the nearest location of the underway surface $pCO_2$
measurements. We limited the observational data from 2001-01-01 to 2014-12-31 to assure the ideal coverage of most
products. Model R by Xue et al. (2016) only had a temporal coverage from 2005 to 2010, underway $pCO_2$ observation data
are compared with model results of the nearest year in Fig. 15. The statistics of model-data comparison are listed in Table 2.
Several global models substantially overestimated the surface $pCO_2$ in the GoM, including B (CanESM5), L (MPI-ESM1.2-
HAM), and M (MPI-ESM1.2-HR), and N (MPI-ESM1.2-LR), O (NorCPM1). This can be observed in Fig. 15 (b,l-o), where
substantial overestimation along the cruise lines (in pink) is present, and also from the Mean Bias of these models listed in
Table 2.

For regional models, the 12-month ML-based climatology product by Chen et al. (2019), model T, has the best performance
in terms of RMSE (35.67) and *skill* (-0.11) (Table 2.). However, the 12-monthly climatology product suffers from a temporal
disadvantage when compared with multi-year monthly climatology and model products with smaller time frequency. For
example, Model T had an R-value of 0.54, while multiyear monthly climatology U had a slightly higher R-value of 0.55, and
the daily-averaged model (this work) had an R-value of 0.59. The multi-year monthly product by Xue et al. (2016), model R,
has the largest RMSE (84.92) among the tested products and overestimates shelf regions while underestimating $pCO_2$ in the
open ocean region (especially the loop current). Overall, model R performs poorly in regard to surface $pCO_2$ with a low R-
value of 0.20 and a low *TSS* of 0.24, and the model in this study can reproduce 80% more variance than that already described
in model R (*skill* of this work over model R is 0.80). In addition, the *TSS* of this work is 0.63, which is higher than that of the
model R (0.24), supporting one of the major findings of this work that the NGoM is a carbon sink instead of a source during
summer. Likewise, the open ocean should not be as strong a carbon sink as Xue et al. (2016) suggested since their estimated
$pCO_2$ on the open ocean is significantly lower than the observations. Model S by Gomez et al. (2020) had a relatively low
RMSE of 42.65, a relatively high *TSS* of 0.57 among all models, and when using model S as the reference, this work generated
a *skill* score of 0.22. Fig. 15(s) revealed that model S tends to overestimate $pCO_2$ on the northwest shelf of GoM and
underestimate the open ocean, especially the southern GoM connected with the Caribbean seas.

In Fig. 16, we extracted the monthly surface $pCO_2$ trend at two buoy locations from the models and climatology products to be compared with the monthly averaged buoy $pCO_2$ measurements. Taylor diagrams are shown for integrated evaluation of the standard deviation and correlation coefficient of each model/climatology product concerning observation at the two buoy locations. At the two buoy locations, most products tend to overestimate the summer $pCO_2$, with model R yielding the largest overestimation. The coastal buoys recorded a typical low during the May, Jun, and July period, but most products failed to capture such a trend. Global models with coarse resolution and simplified, if any, river flux prescriptions generally perform poorly in the coastal region (Fig. 16). Even for the relatively well-performed regional model S and ML-based model T, an overestimation as large as 100 µatm during May or June is found. Such an overestimation likely results in a reduced air-sea $CO_2$ flux when the ocean is a carbon sink (flux estimates see Table 3). As shown in Figs. 16 (a,b), this work can capture the monthly climatology of surface $pCO_2$ at two buoy locations relatively well. Such agreements result in relatively better Correlation Coefficients (with smaller p-values), and Standard Deviation within the range of [25 50] for CoastalMS and [50 75] for CoastalLA in Fig16 (c,d). Both the monthly time series and the Taylor diagrams in Fig. 16 reveal the benefits of this regional model as a good description of the coastal carbon system under the influence of the MARS.

**Table 2.** Statistics of surface $pCO_2$ comparison

| | A | B | C | D | E | F | G | H | I | J | K | L |
|---|---|---|---|---|---|---|---|---|---|---|---|---|
| Mean Bias | 8.8 | 19.25 | 3.71 | -1.81 | 4.96 | -0.57 | 5.06 | 4.33 | -4.01 | 2.69 | 3.54 | 13.36 |
| RMSE | 43.61 | 49.82 | 37.31 | 36.13 | 37.75 | 37.91 | 39.16 | 52.09 | 41.29 | 47.24 | 47.4 | 43.75 |
| R-value (p-value < 1e[-5]) | 0.37 | 0.42 | 0.54 | 0.55 | 0.52 | 0.51 | 0.49 | 0.36 | 0.48 | 0.31 | 0.29 | 0.47 |
| *TSS* | 0.43 | 0.5 | 0.55 | 0.53 | 0.52 | 0.52 | 0.51 | 0.45 | 0.54 | 0.42 | 0.41 | 0.53 |
| *skill* | 0.26 | 0.43 | -0.02 | -0.08 | 0.01 | 0.02 | 0.08 | 0.48 | 0.17 | 0.37 | 0.37 | 0.26 |
| | M | N | O | P | Q | R | S | T | U | V | W | This work |
| Mean Bias | 20.88 | 15.79 | 13.06 | -1.81 | -5 | 0.15 | -5.81 | -1.04 | 3.78 | -2.25 | 1.43 | 0.53 |
| RMSE | 48.96 | 47.28 | 41.35 | 39.2 | 39.55 | 84.92 | 42.65 | 35.67 | 35.93 | 38.77 | 38.9 | 37.62 |
| R-value (p-value < 1e[-5]) | 0.46 | 0.40 | 0.42 | 0.46 | 0.44 | 0.20 | 0.51 | 0.54 | 0.55 | 0.42 | 0.43 | 0.59 |
| *TSS* | 0.53 | 0.49 | 0.38 | 0.48 | 0.44 | 0.24 | 0.57 | 0.42 | 0.34 | 0.32 | 0.38 | 0.63 |
| *skill* | 0.41 | 0.37 | 0.17 | 0.08 | 0.10 | 0.80 | 0.22 | -0.11 | -0.10 | 0.06 | 0.06 | 0 |

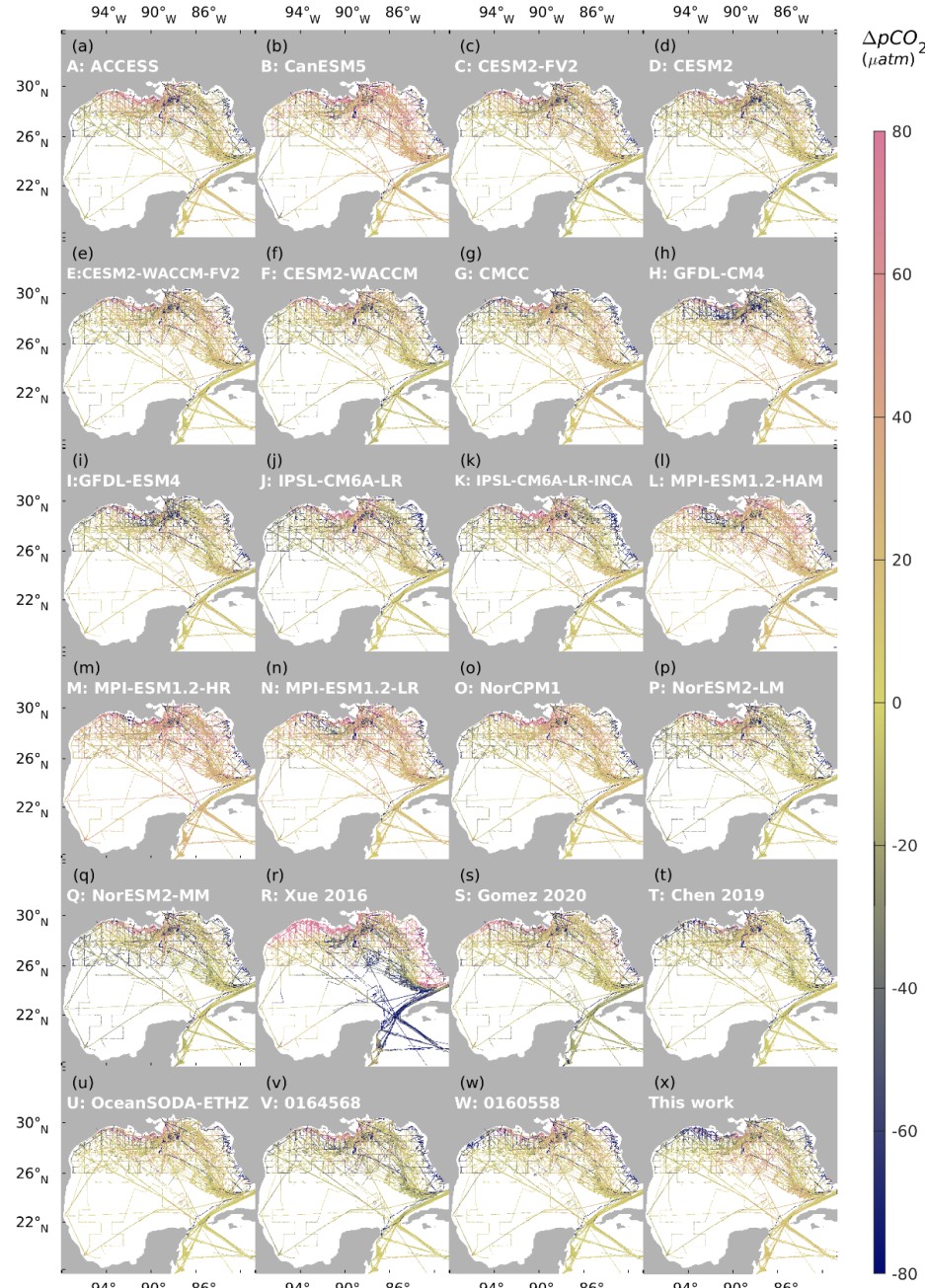

**Figure 15: Comparison of sea surface $pCO_2$ between Global Climate Models (GCMs) (A through Q), global ocean climatology products (U, V, W), regional ocean model products (R, S, T, This work), and underway sea surface $pCO_2$ measurements. A Positive $\Delta pCO_2$ indicates the product data overestimate sea surface $pCO_2$. A negative $\Delta pCO_2$ suggests the product data underestimate sea surface $pCO_2$. A neutral $\Delta pCO_2$ indicates the product data agree well with the observed sea surface $pCO_2$. The white spaces between the cruise lines indicate these regions do not have observational $pCO_2$ data.**

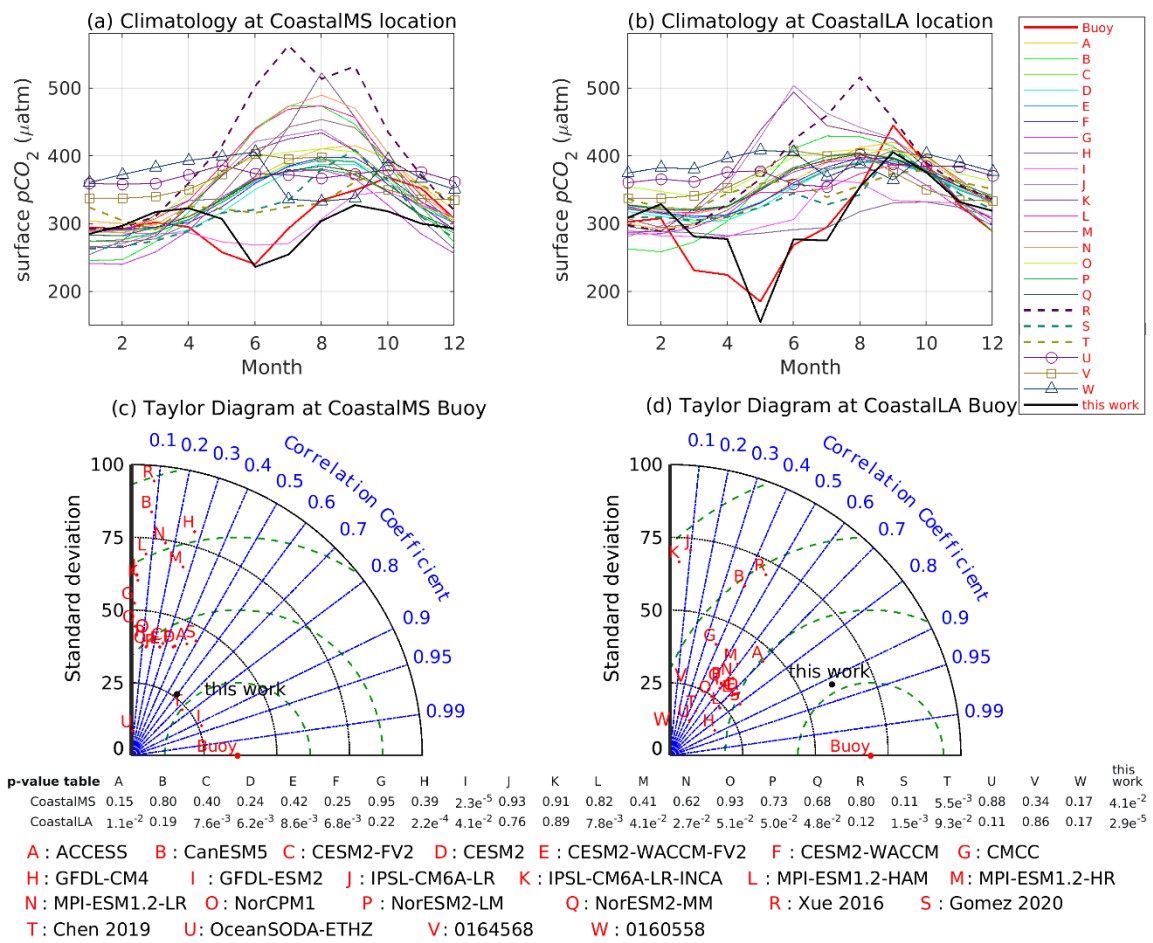

Figure 16. Comparison of sea surface $pCO_2$ from Global Climate Models (GCMs) (A through Q), global ocean climatology products (U, V, W), and regional ocean model products (R, S, T, this work) at two buoy sites. Climatology at the two buoy locations of CMIP6 models and Gomez et al. (2020) are calculated by multiyear averaging from 2000-2014 model surface results. Climatology at the two buoy locations of Xue et al. (2016) is calculated by multiyear averaging from 2005 to 2010. Climatology at the two buoy locations of Chen et al. (2019) is calculated from their 12-monthly ML surface $pCO_2$ product (from 2002-07 to 2017-12). Climatology from NCEI Accession 0164568 and NCEI Accession 0160558 contained 12-month estimates of surface $pCO_2$ and was used directly. The monthly climatology of OceanSODA-ETHZ and this work are calculated from 2017-07 to 2019-12 for CoastalLA buoy and from 2011-03 to 2017-05 for CoastalMS buoy. Buoy raw observations have a frequency of ~ 3 hours, and monthly averages are used to be compared with monthly model estimates. The p-value for each correlation coefficient is listed in the p-value table.

## 5.2 Air-Sea Flux

In Table 3, we compare the annual air-sea $CO_2$ flux generated by this work with that reported by Xue et al. (2016) for 2005–2010, Gomez et al. (2020) for 2005–2014, Robbins et al. (2014) for 1996-2012, Huang et al. (2015) for 2004-2008, and Lohrenz et al. (2018) for 2006-2010. Using the same gas transfer velocity parameterization as this study, Xue et al. (2016) simulated a smaller carbon sink in the NGoM and a larger carbon sink estimation in the Open GoM due to their overestimation of shelf $pCO_2$ and underestimation of the Open GoM $pCO_2$, as shown in Fig. 15(r). The large bias of the Open GoM carbon sink by

Xue et al. (2016) likely results from the over-simplified prescription of the initial and boundary condition of DIC and TA (based on the empirical relationship with temperature and salinity), which led to an overestimation of carbon sink in the Open GoM (1.6 times of the value reported by this work and up to three times of that reported by Gomez et al., 2020). To compare with the flux estimates of Gomez et al. (2020), we rescale our estimates to the gas transfer velocity parameterization used in their work (based on Wanninkhof, 2014) and produced mean estimations of $1.59 \pm 2.13$, $0.52 \pm 0.34$, $0.50 \pm 0.86$ mol m$^{-2}$ yr$^{-1}$ for the NGoM, Open GoM, and Gulf-wide, respectively. It is expected that this work estimated a larger carbon sink in the NGoM region compared with that of Xue et al., 2016 and Gomez et al., 2020, as revealed in Fig. 15(r,s) and Fig. 16(a,b). Surface $pCO_2$ in the coastal region simulated by Xue et al. (2016) is significantly overestimated, as shown in Fig. 15(r) and Fig. 16 (a,b). This model bias corresponds to the smallest NGoM carbon sink estimation in Table 3. Similarly, surface $pCO_2$ in the coastal NGoM region and at the two buoys locations was overestimated by Gomez et al. (2020) (Fig. 15(s), Fig.16 (a,b)). This $pCO_2$ overestimation can be a reason for its smaller carbon sink estimation for the NGoM compared with that of Lohrenz et al. (2018), Huang et al. (2015), and this work. Additionally, the NGoM is a carbon source from June to October according to Gomez et al. (2020), which is different from what we simulated in this study (NGoM is a carbon sink all year round). Combining information from Fig. 16, where model S overestimates $pCO_2$ by $\sim 50$ µatm on average during June at the two coastal locations, we conclude that the NGoM air-sea $CO_2$ sink by Gomez et al. (2020) is likely underestimated. The observation-based studies by Huang et al. (2015) yielded an annual sink of $0.96 \pm 3.7$ mol C m$^{-2}$ yr$^{-1}$ for NGoM, based on the monthly satellite products QuikSCAT wind data (12.5 km resolution). Lohrenz et al. (2018) estimated an annual sink of $1.1 \pm 0.3$ mol C m$^{-2}$ yr$^{-1}$ for NGoM, using gas transfer velocities estimated for each 8-day period. To sum up, we conclude that the air-sea $CO_2$ flux generated by this work is more robust in the NGoM region than that in previous model/climatology products. Nevertheless, a direct carbon flux comparison between model and observation-based studies needs to account for the differences in wind data and the gas transfer velocities.

**Table 3.** Air-sea $CO_2$ flux comparison among this work and previous studies for the GoM. The mean estimate is followed by the Standard Deviation with the $\pm$ symbol. Positive flux indicates the ocean is a carbon sink with regard to the atmosphere.

| Study Type | | | NGoM | Open GoM | Gulf wide |
|---|---|---|---|---|---|
| | | | **mmol m$^{-2}$ d$^{-1}$** | | |
| | | Spring | $4.93 \pm 10.55$ | $2.91 \pm 1.35$ | $2.48 \pm 3.75$ |
| | | Summer | $1.71 \pm 6.19$ | $-1.83 \pm 0.42$ | $-1.55 \pm 2.25$ |
| Model based | This work | Fall | $4.79 \pm 4.93$ | $0.17 \pm 1.04$ | $0.45 \pm 2.47$ |
| | | Winter | $10.00 \pm 9.50$ | $5.80 \pm 2.24$ | $5.40 \pm 4.30$ |
| | | | **mol m$^{-2}$ yr$^{-1}$** | | |
| | | Annual | $1.96 \pm 2.63$ | $0.64 \pm 0.42$ | $0.62 \pm 1.06$ |
| | Xue et al., 2016 | Annual | $0.32 \pm 0.74$ | $1.04 \pm 0.46$ | $0.71 \pm 0.54$ |
| | Gomez et al., 2020 | Annual | $0.93 \pm 1.65$ | $0.33 \pm 0.87$ | $0.35 \pm 1.01$ |
| | Robbins et al., 2014 | Annual | $0.44 \pm 0.37$ | $0.48 \pm 0.07$ | $0.19 \pm 0.08$ |

| Observation based | Huang et al., 2015 | Annual | 0.95±3.7 |
| | Lohrenz et al., 2018 | Annual | 1.1±0.3 |

### 5.3 Outlook and Future Model Development

A likely warmer climate combined with heavier precipitation and greater river discharge is predicted in the following years
for the MARS (Dai et al., 2020; Fischer & Knutti, 2015; Frei et al., 1998; Tao et al., 2014), although climate change might reduce precipitation for some low and middle latitudes regions (Arora & Boer, 2001; Na, Fu, & Kodama, 2020). A warmer climate will reduce the momentum of the Loop Current, and less tropic water (reduced by about 20–25%) will be introduced into the GoM from the Caribbean (Liu et al., 2012). As a consequence, Loop Current might penetrate less into the NGoM and reduce the upwelling along the NGoM and WF slope. Stronger river discharge with nutrient loads will exacerbate the NGoM
acidification in bottom water (Laurent et al., 2018) while increasing the surface water biological $CO_2$ utilization and removal, creating larger river plume regions that exhibit a distinct carbon footprint compared to its surrounding waters. Such predictions resemble the perturbation prescribed in the Bry and NoR experiments, where reduced global ocean impact can be assessed by the difference between Bry and His experiments, and impacts from increased river discharge can be assessed by the difference between His and NoR experiments. We anticipate the Open GoM to be a stronger carbon sink in the future under the projection
of Loop Current weakening. And the NGoM will continue to be a strong carbon sink with the sink region expanded in response to predicted greater river discharge and smaller momentum in Loop Current.

Field samples of the carbon system give us synoptic knowledge of the carbon cycle in the ocean. However, carbon system attributes are subject to large fluctuation due to temperature, salinity, mixing, and biological activities, current observations of
the carbon system at the sea surface or vertically along transects are far from enough to reveal the carbon system evolution in the GoM. As ship-based observations are limited by spatial coverage and temporal coverage, mooring observations have a high-frequency (~3 h) in time but only cover limited geological locations. ML model derived from remote sensing and cruise data inherit the bias from satellite ocean color products and ship-based measurements, and more importantly, it assumes the training data contains all information that defines the system it is trying to predict, which is not necessarily the truth. One
benefit of the numerical models is to offer information to bridge fragmentary knowledge and fill in the gaps between observation and reality. However, the marine carbon cycle is admittedly a complex process. Several simplifications and parameterizations are needed to perform a numerical simulation. Nevertheless, specifications for some key processes may warrant further investigation and better parameterization: 1) The multiple alkalinity generation processes in the sediment pool in current experiments are linked linearly with the aerobic decomposition of PON with a fixed ratio, which can potentially
induce large bias during high PON concentration. The anoxic zone chemistry component can be added to properly simulate the carbonate system in oxygen-deficient conditions (Raven, Keil, & Webb, 2021), which can prevail in bottom boundary layer waters in coastal regions in NGoM, especially during summer. Adding in anoxic zone chemistry will also allow a more

diversified prescription for TA generation, which plays a key role in the understanding of sediment *pH* dynamics (Gustafsson et al., 2019; Middelburg, Soetaert & Hagens, 2020). 2) In our model, the density-related fragmentation/flocculation of detritus OM is simplified with one particulate and one dissolved pool, each with a fixed sinking rate. Coagulation and flocculation can transform DOC into particulate OM, or subsequently form large aggregates, whose remineralization rate can be much faster (Ploug et al., 1999). The remineralization-sinking dynamic determined the fate of OM decomposition (and water column DIC profile) and should be allowed to have more degree of freedom in future model development. 3) Calcification in this work can reflect the primary factors regulating marine calcification. However, important feedback from water acidity on the calcification is omitted due to the overall supersaturation with aragonite in GoM shelf waters. Therefore, the modeled $CaCO_3$/PON ratio could not reflect the decreasing trends of the $CaCO_3$/PON ratio under acidification (Zondervan et al., 2001). 4) Phytoplankton groups can play different roles in carbon cycling given their different sizes, sinking rates, calcification rates, etc., and their relative ratio would be critical to the carbon dynamic (Le Moigne et al., 2015; Poulton et al., 2007). The interplay between zooplankton grazing and phytoplankton bloom in this work captured the seasonal dynamic but only had fixed modes toward nutrient levels. High nutrient concentration favors the success of diatom, and lower nutrient level gives small phytoplankton a competitive advantage in NGoM (Aké-Castillo & Vázquez, 2008; Chakraborty & Lohrenz, 2015; Qian et al., 2003; Strom & Strom, 1996). More phytoplankton groups and possible predation avoidance mechanisms could be added to the model to give the bloom pattern (and subsequently the carbon export) more variance (Liszka, 2018; Rost & Riebesell, 2004). 5) Adding the higher trophic level biology could be the next step to improving the model. Marine fishes are reported to produce precipitated carbonates within their intestines at high rates and contribute to TA increase in the top 1000 meters of ocean waters (Wilson et al., 2009). 6) This model did not include sediment silicate weathering and carbon flux through atmospheric deposition, which can potentially be important sources/sinks of carbon to the ocean waters as well (Jurado et al., 2008; Wallmann et al., 2008).

**6 Conclusions**

This study presents a high-resolution regional carbon model for the GoM, with fully coupled carbonate-chemistry calculations and air-sea interaction. The model can reliably simulate the spatial and temporal pattern of the surface ocean carbon system. We show, for the first time, a solid validation of a regional carbon model via direct comparison against high-frequency $CO_2$ buoys, TA/DIC vertical profiles along the coastal transects, remote sensing-based ML model product, and underway *pCO₂* measurements (surface). We calculated the decadal trends of important carbon system variables such as *pCO₂*, *pH*, air-sea $CO_2$ exchange, and *Ω* over the NGoM and Open GoM regions.

The GoM surface *pCO₂* values experience a steady increase from 2001 to 2019, with an increasing rate of 1.61 µatm yr$^{-1}$ in NGoM and 1.66 µatm yr$^{-1}$ in Open GoM, respectively. Correspondingly, the ocean surface *pH* is declining at a rate of 0.0020 yr$^{-1}$ and 0.0015 yr$^{-1}$ for NGoM and Open GoM, respectively. The surface *Ω* over the NGoM and Open GoM region remains

supersaturated with aragonite during the time span of the model but with a slightly decreasing trend. The carbon sink of both NGoM and Open GoM regions exhibits increasing trends and will continue to increase at a faster pace in the coming years under the prospect of climate change and the rising atmospheric $pCO_2$.

We decouple the influence on surface $pCO_2$ into thermal and nonthermal components and further analyze the surface $pCO_2$
changes due to gross primary production and air-sea $CO_2$ flux. We find that the low temperature during winter and the biological uptake during spring and summer are the primary drivers making GoM an overall $CO_2$ sink. During the modeled period of 2001-2009, the GoM overall is a $CO_2$ sink with a mean flux rate of 0.62 mol C m$^{-2}$ yr$^{-1}$ (11.77 Tg C yr$^{-1}$). The NGoM region is a $CO_2$ sink year-round and is very susceptible to changes in river forcing. The Open GoM region is dominated by thermal effects and converts from carbon sink to a source during summer.


Historical simulation (His) and perturbed tests (Bry, NoR) are performed to determine whether observed changes in the GoM carbon system are driven by secondary effects of carbon accumulation from the global ocean or local forcing, such as river inputs. The results show that, in addition to the increasing atmospheric $pCO_2$ over the GoM, the spatial distribution and trend in carbon system variables could only be explained when the effects of carbon accumulation via boundary conditions and the
impact from river discharge are included. Although eliminating carbon accumulation via boundary in the Bry experiment did not bring a significant difference in surface carbon variables compared with that of His, a clear chemical equilibrium shift between [$CO_3^{2-}$] and [$HCO_3^-$] can be observed at subsurface depths under the perturbation of the accumulative boundary carbon concentrations. With a projected warming climate, we anticipate the GoM to be a stronger carbon sink due to an elevated river discharge and reduced impact from the global ocean.




**Appendix A**

Table A1. Summary of CMIP6 GCMs considered for boundaries conditions of the regional model

| Model Name | Institution* | Resolution (m) latitudinal × longitudinal | DIC | TA | NH4 | NO3 |
|---|---|---|---|---|---|---|
| CESM2 | NCAR | 54137×111951 | available | available | available | not available |
| CESM2-FV2 | NCAR | 54137× 111951 | available | available | available | available |
| CESM2-WACCM | NCAR | 54137×111951 | available | available | available | not available |
| CESM2-WACCM-FV2 | NCAR | 54137×111951 | available | available | available | available |
| MPI-ESM1-2-LR | MPI | 124664×124667 | available | available | available | available |
| MPI-ESM1-2-HR | MPI | 33395×42614 | available | available | not available | available |
| MPI-ESM-1-2-HAM | HAMMOZ-Consortium | 124664×124667 | available | available | available | available |
| ACCESS-ESM1-5 | CSIRO | 109095× 99669 | available | available | available | available |
| CMCC-ESM2 | CMCC | 97659×100093 | available | available | available | available |
| CanESM5 | CCCma | 97659×100093 | available | available | available | available |
| IPSL-CM6A-LR | IPSL | 97659×100093 | available | available | available | available |
| IPSL-CM6A-LR-INCA | IPSL | 97659×100093 | available | available | available | available |
| GFDL-CM4 | GFDL | 110769×99690 | available | available | not available | not available |
| GFDL-ESM4 | GFDL | 110804×99690 | available | available | available | available |
| NorESM2-MM | NCC | 93221×99757 | not available | not available | not available | not available |
| NorESM2-LM | NCC | 93221×99757 | not available | not available | not available | not available |
| NorCPM1 | NCC | 54137×111951 | not available | not available | not available | not available |

* Full name of Institutions:
CCCma: Canadian Centre for Climate Modelling and Analysis (Canada)
CSIRO: Commonwealth Scientific and Industrial Research Organization and Bureau of Meteorology (Australia)
CMCC: Centro Euro-Mediterraneo per I Cambiamenti Climatici(Italy)
IPSL: L'Institut Pierre-Simon Laplace(France)
MPI: Max Planck Institute for Meteorology (Germany)
NCC: Norwegian Climate Centre (Norway)
NCAR: National Center for Atmospheric Research (US)
GFDL: Geophysical Fluid Dynamics Laboratory (US)

Table A2. Model Boundary Frequency

| Boundary Variable | Data Source | Frequency used |
|---|---|---|
| u, v, ubar, vbar, zeta, temp,salt | HYCOM | daily |
| NO3, NH4, PO4, Si(OH)4, DIC, TA, diatom, small phytoplankton, microzooplankton, mesozooplankton, Pzooplankton, CalC, DOC | CESM2-WACCM-FV2 | monthly |
| Oxygen | WOA | static climatology |
| DON, PON, opal | small positive value | constant |

**Code/Data availability:**


**Author contribution:** Z. George Xue designed the experiments and Le Zhang carried them out. Le Zhang developed the model code and performed the simulations. Le Zhang and Z. George Xue prepared the manuscript.

**Competing interests:** The authors declare that they have no conflict of interest.


**Acknowledgment:** Research support provided through the National Science Foundation (award number OCE-1635837; EnvS 1903340; OCE- 2049047; OCE-2054935), NASA (award number NNH17ZHA002C), Louisiana Board of Regents (award number NASA/LEQSF (2018-20)-Phase3-11), NOAA Graduate Research Fellowship in Ocean, Coastal and Estuarine Acidification (OA R/CWQ-11). Computational support was provided by the High-Performance Computing Facility (clusters

SuperMIC and QueenBee3) at Louisiana State University. Calculated $CO_2$ flux data is available at the LSU mass storage system and details are on the webpage of the Coupled Ocean Modeling Group at LSU (http://coastandenvironment.lsu.edu/docs/faculty/xuelab/). Data requests can be sent to the corresponding author via this webpage.

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
