# Peer review of "A Numerical Reassessment of the Gulf of Mexico Carbon System in Connection with the Mississippi River and Global Ocean"

_Biogeosciences, 2021_

## Author Comment (AC1)

**Thanks for the comments. We are carrying out a thorough revision addressing these comments.**

**RC 1:** Global climate models are of questionable utility in many regions due to poor spatial resolution and a poor reproduction of riverine inputs and other critical determinants of biogeochemical processes. Downscaling approaches are therefore critical in many regions. Zhang and Zhu present a new "downscaling" of CMIP6 model output for the region surrounding the Gulf of Mexico, and they draw conclusions about recent changes in the region's carbon dynamics. The model used by Zhang and Zhu appears equally or more robust than prior models of the regional carbon budget. This is therefore potentially interesting and relevant work. However, in its present form the manuscript is needlessly confusing and misleading and features some potentially major methodological issues. I therefore recommend that the authors carry out a thorough revision of the manuscript text and to clarify methodological issues. The core contribution of this study is to provide updated (and potentially more robust) estimates of carbon fluxes in this region and to estimate temporal trends in variables such as pCO2 and pH. This is a valuable contribution to the literature as these values continue to have high uncertainties, and I hope the authors can address the concerns below. It is highly misleading to call this a "downscaling" of a CMIP6 model. At present, the title, abstract and introduction misrepresent the work in the paper. The title of the manuscript claims this study downscales the global CESM2-WACCM-FV2 model. Conventionally, this should mean that all possible driving data is derived from the global model. Critically, any climate forcings should come from the global model. However, as stated on page 7 of the manuscript, the only things taken from the CESM2-WACCM-FV2 model are the initial conditions and boundary conditions on the geographic boundary. Atmospheric forcings etc. are not taken from the CESM2-WACCM-FV2 model. I therefore view this as a hindcast, where the authors were forced to use the CESM2-WACCM-FV2 model for geographic boundary conditions as a compromise. In no real sense is it a downscaling of a CMIP6 model. This is a major problem for the paper as there are, at present, many inaccurate statements. For example, the abstract claims this: "The model's biogeochemical cycle is driven by the Coupled Model Intercomparison Project 6-Community Earth System Model 2 products (CMIP6-CESM2)..." This is clearly not true, as surface temperature, air PCO2, riverine inputs and most of the variables driving the carbon dynamics do not come from the CMIP6 product. The title, and aims of the paper should therefore be revised. The paper really appears to be a new estimate of carbon fluxes in the region. It should therefore be rewritten accordingly. Critically, the authors should make it clearer how, as claimed, the estimates in this study are more reliable than previous methods. The evidence provided for this are not extensive.

**Response:** We acknowledge that the term "downscaling" might not be appropriate, as suggested by reviewers #1 and #2. In this revision, we feature this paper as a hindcast that provided a new estimate of carbon fluxes of the Gulf of Mexico (GoM) and filled the current knowledge gap of the available carbon monitoring data in the GoM. The accuracy of carbon flux estimation in a regional ocean like the GoM is still limited by surface  $pCO_2$  data availability. Our model study, to our knowledge, is the first one in this region to use global products as the boundary condition for biogeochemical fields. Via extensive model-data comparison against three data sources: high-frequency *in situ* buoy measurement, machine learning product based on remote-sensing and underway  $pCO_2$  observations, and CTD transects, our study demonstrates that it is feasible to utilize historical run of the global products to drive a regional coupled physical-biogeochemical model. We will revise all relevant statements. We intend to change the title into "A Re-assessment of the Gulf of Mexico (GoM) Carbon System: Connecting the Gulf of Mexico with the Mississippi River and the Global Ocean".

**RC 1**: *Output of the CESM2-WACCM-FV2 model are used for both initial and boundary conditions. The authors do not state why they used the CESM2-WACCM-FV2 model for the boundary conditions. Was*

this model more accurate in the region than other CMIP6 models or reanalysis products that are available? This is a critical question, as it is possible the choice has reduced the reliability of the carbon budget estimates. There are also specific issues surrounding the use of this dataset. First, this model can have negative values for nitrate, and presumably other variables. I viewed one of the historical files (http://esgf-data.ucar.edu/thredds/fileServer/esg\_dataroot/CMIP6/CMIP/NCAR/CESM2-WACCM-FV2/historical/r1i1p1f1/Omon/no3os/gn/v20191120/no3os Omon CESM2-WACCM-*FV2* historical r11p1f1 gn 200001-201412.nc) for this model and negative values for nitrate appear very frequently across the boundary. Translating these values into boundary conditions is not a trivial issue as mass conservation etc. is ambiguous. The authors need to explain this thoroughly. Negatives at the boundary also result in average conditions that are far lower than those you would get from the NOAA World Ocean Atlas. Potentially this has been corrected for in some way by the authors, but if it has not it is not clear if the treatment of the boundary conditions is sensible. Likewise, there are negative values in the first time step in 2000, which the authors presumably used in some way to generate their initial conditions. The authors state on p. 19 that this study's estimates of air-sea CO2 fluxes are "more reliable than previous GoM model studies". However, without showing whether the boundary conditions are reliable it is difficult to assess this claim. This is especially true, given the authors state that Xue et al. 2016 used over-simplified boundary conditions. There is therefore real potential that the boundary conditions used here are no more reliable than those in Xue et al.

**Response:** We did a thorough examination of available global ocean carbon-related climatology/ reanalysis products for the potential to be used as the regional model boundaries. And we argue that the current model boundaries setting is a better choice. It should be noted that the GoM region has very limited observations of dissolved inorganic carbon (DIC) and total alkalinity (TA) available, and observational data along different depths are even fewer and has limited spatial and temporal coverage. This is the primary reason many global climatology products have no coverage in the GoM region, e.g., Mapped Observation-Based Oceanic DIC monthly climatology from the Max-Planck-Institute for Meteorology (MOBO-DIC MPIM) (NCEI Accession 0221526) (Keppler et al. 2020); or only contain surface carbon variables, e.g., global gridded data set of the surface ocean carbonate system OceanSODA-ETHZ (v2021,NCEI Accession 0220059) (Gregor and Gruber, 2020), Climatological Distributions of pH, pCO2, Total CO2, Alkalinity, and CaCO3 Saturation in the Global Surface Ocean (NCEI Accession 0164568) (Takahashi et al. 2017), the partial pressure of carbon dioxide collected from Surface underway observations in the world-wide oceans (NCEI Accession 0161129) (Bakker et al. 2017), an observationbased global monthly gridded sea surface  $pCO_2$  product (NCEI Accession 0160558) (Landschützer et al., 2017), and a global ocean  $pCO_2$  climatology combining open ocean and coastal areas (NCEI Accession 0209633) (Landschützer et al. 2020).

Thus global ocean climatology products are not suitable to be used as the boundary for a regional model since data along the vertical direction are needed for the model boundary. The most updated global monthly climatology of TA (NCEI Accession 0222470) (Broullón et al. 2020b) and DIC (NCEI Accession 0222469) (Broullón et al. 2020a) offer 12-month climatology with a 1°x1° spatial resolution and 102 depth levels, this product can potentially be used as a static DIC, TA boundary for the regional model. However, these products utilized a neural network approach to achieving full data coverage for the 3-dimensional global ocean. They used  $pCO_2$  from LDEOv2016 (Takahashi, Sutherland, Kozyr, 2017) and TA from Broullón et al. (2019) to compute DIC surface values to increase the surface coverage in the training data for the machine learning model (Broullón et al. 2020a,b). Therefore, we should be aware that the generated global monthly climatology products are not purely observation interpolations but rather a machine learning model product with many untested assumptions. And the calculated DIC values do not necessarily match the field observations even if the temperature, salinity, TA, and  $pCO_2$  source data used for the calculations are accurate. Although the global climatology products include grided estimates in the GoM region over 12 months, the products are ultimately derived from limited observations that cannot

support such variation in space and time without data augmentation. We argue that the GCMs based on biogeochemical processes, earth system circulations, and conservation schemes are more reliable than the neural network machine learning model used to generate the climatology product.

CMIP6 participating GCMs consume enormous research resources and generate unprecedented knowledge on global carbon system evolution with a whole-ecosystem conservation perspective. Utilizing GCMs results in a refined regional model extends their research value, especially in regards to bridging GCMs product with *in situ* field observations. With the interannual variation estimated by GCMs, the regional model should take advantage of global models by using dynamic boundaries that reflect climate oscillations and carbon accumulation in oceanic waters. The choice of CESM2-WACCM-FV2 model among other global climate models is primarily based on the global model's horizontal resolution in the GoM region and the availability of nutrients and carbon variables. By comparing a number of global models (Table A1), we conclude that the CESM2-WACCM-FV2 by NCAR is among the best modeling resolution in the Gulf of Mexico region- it contains all essential nutrients and carbon variables available and has a natural focus on the US waters.

We can use a static climatology product boundary or like Xue et al. (2016) use an empirical salinity– temperature–DIC–alkalinity relationships to prescribe the model DIC and TA boundaries. However, using a static boundary or an over-simplified boundary like Xue et al. (2016) would reduce the value of the current study. As shown in Fig. A1 the World Ocean Atlas nitrate product (Boyer et al. 2018) not only does not provide a dynamic boundary with interannual variability but also contains very limited supporting observational data at the domain boundaries, both horizontal and vertically. Therefore we would refrain from using WOA or other climatology as the regional model boundaries.

Figure A1. World Ocean Atlas 2018 NO3 product mean fields (a) surface, (c) bottom and all available observation counts (b) surface, (d) bottom incorporated in the product. (note: for observation counts below 10, a single-digit number is shown; for observation counts > 100 same color grade is shown as that of 100)

| Model Name            | Institution           | Resolution (m) | NH4              | NO3           |
|-----------------------|-----------------------|----------------|------------------|---------------|
| CESM2                 | NCAR                  | 124214.044     | available        | not available |
| CESM2-FV2             | NCAR                  | 124214.044     | available        | available     |
| CESM2-WACCM           | NCAR                  | 124214.044     | available        | not available |
| CESM2-WACCM-
FV2   | NCAR                  | 124214.044     | available        | available     |
| MPI-ESM1-2-LR         | MPI                   | 176531.166     | available        | available     |
| MPI-ESM1-2-HR         | MPI                   | 53841.4877     | not
available | available     |
| MPI-ESM-1-2-
HAM   | HAMMOZ-
Consortium | 176531.166     | available        | available     |
| ACCESS-ESM1-5         | CSIRO                 | 147378.349     | available        | available     |
| CMCC-ESM2             | CMCC                  | 140351.946     | available        | available     |
| CanESM5               | CCCma                 | 140351.946     | available        | available     |
| IPSL-CM6A-LR          | IPSL                  | 140352.354     | available        | available     |
| IPSL-CM6A-LR-
INCA | IPSL                  | 140352.354     | available        | available     |

Table A1. Summary of CMIP6 GCMs considered for regional boundaries

Translating negative tracer values into boundary conditions from the global model product is indeed a question to be considered when implementing the regional model. Out-of-bound tracer value is a common occurrence to all numerical modeling and is related to the tracer advection scheme used. Numerical schemes for tracer transport and mixing ideally satisfy high-order accuracy, conservation, and boundedness. However, boundedness is generally not strictly imposed as most numerical schemes give priority to the former two desirable properties. Commonly-used approaches to enforce tracer boundedness either compromise accuracy or conservation. Minor occurrences of out-of-bound tracer value *per se* should not debase the credibility and reliability of GCMs, which have to meet stringent modeling requirements (finer grid resolution and smaller time step can reduce the occurrence of negative tracer values, but balancing the computational cost and model complexity dissuades such implementation). The negative concentration of tracers can be corrected with a mass conservative and non-diffusive scheme by balancing the value from the nearest grid points in a way that conserves the tracer mass. Most negative tracer data for NO3 happens in the middle of loop current where there is a sudden change in bathymetry and loop current takes a sharp turn as a result of the combined stresses. Along the southern boundary of our model (around latitude 16.7387 N), we notice negative tracer values rarely happen (Table A2).

| Model Name        | Institution           | NO3 range in GoM southern boundary    | NO3 range in GoM eastern
boundary                           |  |
|-------------------|-----------------------|---------------------------------------|----------------------------------------------------------------|--|
|                   |                       | [min, max], unit: mol m -3 | [min, max], unit: mol m -3                          |  |
| CESM2-FV2         | NCAR                  | [0, 0.026418]                         | [-0.00033723, 0.026207]
(1.34 % negative tracer
values)  |  |
| CESM2-WACCM-FV2   | NCAR                  | [0, 0.025887]                         | [-0.00025584,
0.025676]( 0.98 % negative
tracer values ) |  |
| MPI-ESM1-2-LR     | MPI                   | [0, 0.023258]                         |                                                                |  |
| MPI-ESM1-2-HR     | MPI                   | [0, 0.020868]                         |                                                                |  |
| MPI-ESM-1-2-HAM   | HAMMOZ-
Consortium | [0, 0.022392]                         |                                                                |  |
| ACCESS-ESM1-5     | CSIRO                 | [0, 0.073462]                         |                                                                |  |
| CMCC-ESM2         | CMCC                  | [0, 0.048223]                         |                                                                |  |
| CanESM5           | CCCma                 | [0, 0.06836]                          |                                                                |  |
| IPSL-CM6A-LR      | IPSL                  | [0, 0.041679]                         |                                                                |  |
| IPSL-CM6A-LR-INCA | IPSL                  | [0, 0.041748]                         |                                                                |  |

 Table A2. GCMs Tracer NO3 value range along GoM boundary

**RC1:** Only a single year is used for model spin up. It is not clear if the model will really have settled down by that point. Many regional models require 5 years to spin up, so one year is possibly questionable, especially given model output is used for temporal trend analysis.Starting conditions are used from the CESM2-WACCM-FV2 model, and quasi-equilibrium conditions for this model will differ (perhaps quite dramatically) from the regional model. The authors justify using a one-year spin up by saying "the global model has been well stabilized up to the year 2000 from its 'pre-industry' experiment". This does not say much about the stability of the regional model used. Given the issues mentioned above about negative nitrate values in the global model, it seems questionable whether the starting conditions are close to a stable state in the regional model. Furthermore, it is plausible that riverine inputs are drastically better resolved in the regional model than the global model. This is particularly important given the conclusion of the importance of the carbon inputs from the Mississippi River.

The spin-up timing issue is also particularly relevant for the "no rivers" experiment. This experiment essentially removes rivers at the start of 2000, but assumes that the model is effectively spun-up to "river-free" conditions by the end of 2000. The authors need to show that this is credible. Otherwise, some of the results in the experiments section may not be robust.

**Response:** The regional model has the benefit of swift spin-up compared with the global model due to higher spatial resolution, smaller time-step, and relatively high momentum in the GoM region. The biogeochemical model typically completes its spin-up in one year (e.g. Große, Fennel, Laurent, 2019;

Laurent and Fennel, 2019; Laurent et al., 2021). To address the reviewer's comment about the spin-up time, especially for the "no rivers" experiment, we added the spin-up comparison results of the "no rivers" experiment to assure that a one-year spin-up is necessary and further spin-up beyond one year do not gain additional benefits. We used the 9-year spin-up result as the control and assumed spinning up for nine years was adequate for the regional model. Then we check the difference between the model results of different spin-up times to that of the 9-year spin-up result (as Diff). For the interest of the carbon model, ocean surface DIC and TA concentration differences are plotted in detail in Figs. A2 and A3.

---

## Author Comment (AC2)

Thanks for these comments! We found the two reviewers shared the same insights in several points; in such cases, we will direct our responses to Reviewer #1 to avoid redundancy.

*RC2: Review of "Downscaling CMIP6 Global Solutions to Regional Ocean Carbon Model: Connecting the Mississippi, Gulf of Mexico, and Global Ocean"*

This paper presents a 20-year simulation of the Gulf of Mexico with a coupled physical biogeochemical model including carbon chemistry. The paper provides a validation of the model against observations. It also presents two perturbed experiments, "Bry" and "NoR", "Bry" has fixed the DIC and TA boundary conditions for the year 2000 and "NoR" has no rivers. Furthermore, the carbon budget and how different processes such as temperature, primary production and mixing attribute to the total is presented, which was very interesting to see. The presentation is focused on the Northern Gulf of Mexico and the open ocean Gulf of Mexico, though some other regions are also discussed. This work is interesting and worthy of publication, but a major revision of how the results are presented is necessary before publication. Here are the major points:

The title promises a downscaling of the Gulf of Mexico using a regional model. I was expecting an actual downscaling in a global model, including a historical simulation and forward projection. However, the presented model appears more like a hindcast forced by NCEP reanalysis with initial and boundary conditions from the climate model rather than a full downscaling. Several questions arise related to this:

Why was this climate model selected?

Why was a climate model selected for the boundary and initial conditions as opposed to either climatology or a global ocean reanalysis?

What is the biogeochemical model that was used in the climate model and is it of a similar complexity to NEMURO?

**Response**: We will correct the usage of the "downscaling" terminology, as suggested by both reviewers. Please refer to our detailed response to Reviewer #1 on selecting the GCMs as the boundary condition among available global climatology/ reanalysis products. The biogeochemical model used in this study is built upon the NEMURO (Kishi et al., 2007), and is of similar complexity. NEMURO model originally configured eleven state variables including nutrients (Si(OH)4, NO3, NH4), plankton groups (ZP: predator zooplankton, ZL: large zooplankton, ZS: small zooplankton, PL: large phytoplankton, PS: small phytoplankton), dissolved organic nitrogen (DON), particulate organic nitrogen (PON), and opal (OPL). We noticed a typo in the manuscript that misspelled eleven as seven. The added carbon cycle is linked with the original nitrogen cycle with a fixed ratio (the Redfield ratio of C: N = 6.625). Table B1 lists each added variable and their referenced studies.

| Variables added to NEMURO model  | Reference                                                                                               |
|----------------------------------|---------------------------------------------------------------------------------------------------------|
| Dissolved Inorganic Carbon (DIC) | Respiration and remineralization linked with the nitrogen cycle;(Moore et al., 2004)                    |
| Total Alkalinity (TA)            | Alkalinity generation in sediment processes (Fennel et al., 2006; Hu & Cai, 2011); (Moore et al., 2004) |
| Dissolved Organic Carbon (DOC)   | (Moore et al., 2004)                                                                                    |
| Calcium Carbonate (CalC)         | Calcification rate (Moore et al., 2004);                                                                |
|                                  | Dissolution (Mucci, 1983; Millero, 1982, 2007,                                                          |

Table B1 Newly Introduced Tracers to NEMURO by this study

| 1995);                                                                              |
|-------------------------------------------------------------------------------------|
| (Moore et al., 2004)                                                                |
| Linked with carbon/ nitrogen stoichiometry in aerobic respiration or photosynthesis |
| Followed oxygen air-sea flux parameterization (Wanninkhof, 1992)                    |
|                                                                                     |

**RC2**: Validation: It is very good that more than one type of observation is used, but the results are presented only graphically. I would like to see some statistical quantities such as bias and rmse. Additionally, the differences could be shown: for example in Figure 5, I would have preferred to see seasons rather than month, but then adding the plots for differences.

An important motivation for doing the study was the improved quality of this downscaling compared to the earth system model, to demonstrate that, also some error estimated from the ESM should be included for comparison.

**Response**: Thanks for the suggestions. Reviewer #1 also has the same advice on adding statistics to model-data comparison. Please refer to our response to this point suggested by Reviewer #1. We plan to add model *skill* (Eq. A1), the Taylor Skil Score (TSS) (Eq. A2), and commonly used statistical metrics such as correlation coefficient (R), Root Mean Square Error (RMSE), and the standard deviation (STD) to evaluate the model performance compared to the GCMs and global climatology products. For Fig. 5, we plan to use seasonal means and add subplots for the difference between the two products. Regarding the comparison of the improved quality of our downscaled model between the global model, please refer to Figure A4 in our response to Reviewer #1.

**RC2:** *Structure: In the results the main run is presented and then discussed against previous estimates. Then in the discussion, results from the two perturbed experiments are presented. In my view the discussion of the main run against other studies belongs in the discussion, while the presentation of the perturbed runs belongs in the results.*

**Response:** The structure issue of mixing results and discussion will be corrected. We will compare with other studies in the discussion section and present the results of the perturbed runs in the result section.

**RC2:** Clarity on analysis: I spent a fair amount of time trying to understand the distinction between the different "types" of CO2 in the analysis as described by equations 4, 5 and 7. There is no equation (6) it seems. First of all, these could all be presented together and some work is needed to make this more understandable. Additionally it would help if in table 3 also the actual pCO2 was presented. I assume the triangular bracket is the temporal mean, but it should be stated. Why compute the contribution from GPP and not NPP? Furthermore the thermal contribution I would understand as "How much higher or lower the pCO2 is because the temperature is either higher or lower than the mean". Here however the number presented is of same order as the actual pCO2. I would also expect that the thermal and the non-thermal part would add up to total pCO2, however adding equation 4 and 5 does not yield pCO2. So better explanation is needed in this part. I find the labeling "mixing" of the last pCO2-term presented in eq. 7 misleading, see my comment further down.

**Response**: All equations concerning different components of  $CO_2$  will be presented together, and  $pCO_2$  thermal will be re-defined to allow a sum of all  $CO_2$  components to add up to the apparent  $CO_2$ . The actual/apparent  $pCO_2$  will be added in Table 3 for better comparison. The contribution from gross primary

production (GPP) is used because this is the process that directly affects the CO2 uptake. Respiration in the ocean water column is much more complex to keep track of than that of photosynthesis. In the model, GPP can be conveniently calculated by tracking the photosynthesis activity of diatom and small phytoplankton (which is a function of solar radiation, temperature, nutrients, and phytoplankton concentrations). Respiration concerns both living biota (phytoplanktons, zooplanktons) and nonliving detritus (particulate organic matter, dissolved organic matter). More importantly, it would be problematic to only account for surface respiration, considering that most detritus sink and respire in deeper water. It seems to be more appropriate to leave the respiration in the end-member of the  $CO_2$  components. In other words, net primary production (NPP) is not a readily calculatable quantity in the model. Because respiration can be allochthonous through advection or sinking (depth-dependent), it can be misleading if incorporated in surface spatial presentation. We used GPP as a component of  $CO_2$  measuring the intensity of photosynthesis, which is a primary driver for surface ocean  $pCO_2$  dynamic in the Gulf of Mexico. The label of the last  $pCO_2$  term was "mixing", which represents various mixing processes (e.g. river water and oceanic water mixing, vertical mixing of upwelled waters, horizontal advection induced lateral transport of tracers with concentration gradients, and entrainment of waters with different chemical nature (i.e. temp/ salt/ DIC/ TA/ detritus concentration)). Remineralization and respiration are included in this term due to the result of the two processes altering water chemical nature (DIC, TA, detritus concentration), and the impact from water chemical nature on  $pCO_2$  is constantly being modified by (and as a result of) the mixing process. We understand the reviewer's concerns since the typical effects from horizontal advection and vertical mixing are distinct enough to be treated separately, especially in the global models. Still, we cannot find a better representation to serve as the label. Also, similar classifications or labels are used in literature (Meléndez et al., 2022; Wanninkhof et al., 2019), and in these studies, horizontal advection is considered as included in the mixing term. As a result, we appeal to keep the "mixing" label.

 $pCO_2$ th was originally defined as the effect of temperature changes on  $pCO_2$ , and the  $pCO_2$ nt was defined to remove the temperature effect from the observed  $pCO_2$  (Takahashi et al., 2002). Numerous literatures followed these definitions (Fay and McKinley, 2017; Landschützer et al., 2018; Lerner et al., 2021; Yao and Hu, 2017).

Although our previous definition of  $pCO_2^{th}$  and  $pCO_2^{nt}$  followed Takahashi et al.(2002), we understand the review's perspective. We would like to re-define the  $pCO_2^{th}$  to make the two parties sum up to the apparent  $pCO_2$ . Therefore we modified the definition of  $pCO_2^{th}$  as equation (B1). And we will modify Fig. 8 with the updated definition and unified color schemes. In equation (B1) and (B2), <SST> denotes the mean SST value over the studied period.

$$pCO_2^{th} = pCO_2 \cdot [1 - exp(\gamma_T \cdot (\langle SST \rangle - SST))]$$
(B1)

$$pCO_2^{nt} = pCO_2 \cdot exp(\gamma_T \cdot (\langle SST \rangle - SST))$$
(B2)

**RC2:** The results are very focused on the surface, it would have been interesting to see more of what goes on below the surface, for example it would be interesting to see the depth of the dissolution horizon for calcite and aragonite.

**Response**: We included vertical transect profiles of DIC, TA validation in our original submission and would like to show more below-surface results. However, there is a lack of measurements for subsurface and especially for water column that is deeper than 200 m. Here we show a vertical profile of calcite and aragonite saturation along the 200m isobath in Figs. B2 and B3. The coastal ocean (

Figure B2. Multiyear averaged Calcite saturation state (2001-2019) along the 200 m isobath (red line in (a)) at the sea bottom in the 150-1000 m depth range (a) top view (b) stretched view. Intersections are chosen along the 200 m isobath line with direction normal to the 200-isobath tangential.

Figure B3. Multiyear averaged Aragonite saturation state (2001-2019) along the 200 m isobath (red line in (a)) at the sea bottom in the 150-1000 m depth range (a) top view (b) stretched view. Intersections are chosen along the 200 m isobath line with direction normal to the 200-isobath tangential.

**RC2:** *Title* "...*Model: Connecting the GoM to the Mississippi and the Global Ocean" would be more correct as this study does not address the influenc of GoM on rivers or the global ocean.*

**Response**: As mentioned above, "Downscaling CMIP6" is misleading as to the nature of the simulations presented. We intend to change the title into "A Re-assessment of the Gulf of Mexico (GoM) Carbon System: Connecting the Gulf of Mexico with the Mississippi River and the Global Ocean".

**RC2**: Line 8: "...reduce uncertainties in spatial..." I do not agree that models reduce uncertainties in estimates, they do however complement observations to fill spatial and temporal gaps in the observation record (with some uncertainty).

Line 20: "confirms": Also write what it confirms, previous models, observations, both? Line 23: Be more specific on how the Mississippi inflow influences the carbon cycle. The last sentence seems obvious as those are the places with inflow to the GoM, but really, when comparing "His" and "Bry", the results are so close to each other I would say a more accurate conclusion from that perturbation experiment is that interannually varying lateral boundary conditions are not necessary on this timescale.

**Response**: we will remove the assertion on "reduce uncertainties", and highlight the contribution to fill spatial and temporal gaps in the observation record. We will articulate what this model confirms – this model confirms with several previous models and ocean surface  $pCO_2$  observations that the riverdominated northern GoM (NGoM) is a substantial carbon sink, and the open GoM is primarily controlled by thermal effect. We will be more specific on how the Mississippi inflow influences the carbon cycle in the revision in three facets, namely the coastal biogeochemical processes, river DIC/DOC/TA budget delivered to the coastal ocean, and atmosphere-ocean-sediment carbon fluxes. Nutrient-fueled primary production removes DIC from surface water and extracts carbon from the atmosphere. Sinking organic matter undergoes decomposition and burial at the ocean bottom, contributing to bottom hypoxia and acidification. Sediment processes remineralize organic carbon, possibly dissolve particulate inorganic carbon, subsequently alter bottom water DIC/TA concentration. The difference between "His" and "Bry" experiments is more than just between multiyear means, which will surely be subtle. Clear differences in interannual carbon system variables due to dynamical boundaries at abnormal years can be observed before such signals are erased in averaging. Due to this reason, we still hold the opinion that interannually varying lateral boundary conditions are necessary for this 20-year model.

**RC2**:*Introduction Line 39 "works" should be "studies"* **Response**: Thanks for the correction, we will change the "work" in Line 39 into "studies".

**RC2: Methods**

Line 120: Be more specific: which variables were originally in NEMURO, which have been added? Line 134 and onwards: could be helpful with a table where each process added is connected to the relevant publication. What is the temporal resolution of the boundary conditions?

**Response**: Variables originally in NEMURO: large phytoplankton, small phytoplankton, predator zooplankton, large zooplankton, small zooplankton, opal, DON, PON, Si(OH)4, NO3, NH4; Added variables: DIC, TA, DOC, CaCO3, PO4, O2. We added Table B1 elaborating where each process is connected with the relevant publication (see Part I). The temporal resolution of the boundary conditions is monthly for ecological variables (DIC, DOC, TA, NO3, Si(OH)4, PO4, NH4) taken from GCM. The temporal resolution of the boundary conditions is static from WOA for O2. The temporal resolution of the boundary conditions is daily for physical variables taken from HYCOM (temperature, salinity, zeta, u, v, ubar, vbar). The temporal resolution of atmospheric forcing is 6-hourly from CFSR, CFSv2 (shortwave radiation, longwave radiation, UWind, Precipitation, air temperature, air pressure, humidity). (see Table A4 in response to Reviewer #1)

**RC2: Validation**

Figure 4: Stretching the y-axis on the upper part of the water-column and putting a black dot in the middle of the observation circle could help to better visualize the difference between the model and observations.

**Response**: Thanks for the detailed suggestions. We will stretch the y-axis on the upper part (

---

## Author Response (AR2)

Dear Editor and Reviewers:

Thank you for the 2nd round comments. We have tried our best to address these comments.

**RC 1:** *The authors have made major revisions to the manuscript based on previous comments by 2 reviewers. Overall, the authors have done a good job of addressing the comments of the reviewers. In particular, they have addressed the issue of the initial paper inaccurately describing the work as a dynamical downscaling of CMIP6 products. I believe the manuscript is almost ready to be published, but the authors should spend some time clarifying some elements.*

*My comments will address a few aspects of the revised manuscript that the authors should address before publication. These comments focus mostly on new material in the manuscript.*

**Response:** We want to thank both reviewers and the associate editor for the invaluable suggestions in the first round and second round review process, and we are clarifying these final elements in this comment.

**RC 1:** *Title*

*The title is more accurate than the previous one. However, the use of the term "Reassessment" leads to an expectation that comparisons with previous studies would be more prominent, especially in the abstract. These comparisons are made in table 3, but they should also be included in the abstract. Overall, the study does show that the new analysis is likely more accurate than previous assessments, but it is probably not as clear as it could be. See further comments below.*

**Response:** To better reflect the title, we added contents on regional model comparison in the abstract – "A reassessment of air-sea CO2 flux with previous modelling and observational studies give us confidence that our model provides a robust and updated CO2 flux estimation, and NGoM is a stronger carbon sink than previously reported."

**RC 1:** *Abstract*

*"The biogeochemical boundaries were interpolated from NCAR's CESM2-WACCM-FV2 solution after a comprehensive evaluation of 17 Global Climate Model (GCMs) products against available observations and global climatology products". This statement is too strong for the abstract. The evaluation of the models is robust enough for this paper, but it is not particularly comprehensive in general.*

**Response:** We adjust the statement to - "The biogeochemical boundaries were interpolated from NCAR's CESM2-WACCM-FV2 solution after evaluating 17 GCMs' performance in the GoM waters."

**RC 1:** *Description of model drivers*

*The description of the model driving data has improved in the latest manuscript. However, it is still somewhat difficult to follow. I recommend that the authors state clearly in a couple of sentences what*

*their overall philosophy is for selecting the driving/boundary data, instead of only stating what data they have chosen. This will address obvious "Why did the study not use X?" questions from readers?*
*The authors should probably also state why they did not use a global biogeochemistry reanalysis product instead of a CMIP6 model. This appears justified, as I believe none of the openly available reanalysis have sufficient variables. But the choice of CMIP6 over a reanalysis may seem questionable to some readers, so clarification could help.*

**Response:** In the previous submission (Line 201) we state, "The choice of CESM2-WACCM-FV2, among other GCMs, is primarily based on its horizontal resolution in the GoM region and its availability of nutrients and carbon variables (see Table A1 for more details)."

Table A1. Summary of CMIP6 GCMs considered for boundaries conditions of the regional model

| Model Name | Institution* | Resolution (m) latitudinal × longitudinal | DIC | TA | NH4 | NO3 |
|---|---|---|---|---|---|---|
| CESM2 | NCAR | 54137×111951 | available | available | available | not available |
| CESM2-FV2 | NCAR | 54137× 111951 | available | available | available | available |
| CESM2-WACCM | NCAR | 54137×111951 | available | available | available | not available |
| CESM2-WACCM-FV2 | NCAR | 54137×111951 | available | available | available | available |
| MPI-ESM1-2-LR | MPI | 124664×124667 | available | available | available | available |
| MPI-ESM1-2-HR | MPI | 33395×42614 | available | available | not available | available |
| MPI-ESM-1-2-HAM | HAMMOZ-Consortium | 124664×124667 | available | available | available | available |
| ACCESS-ESM1-5 | CSIRO | 109095× 99669 | available | available | available | available |
| CMCC-ESM2 | CMCC | 97659×100093 | available | available | available | available |
| CanESM5 | CCCma | 97659×100093 | available | available | available | available |
| IPSL-CM6A-LR | IPSL | 97659×100093 | available | available | available | available |
| IPSL-CM6A-LR-INCA | IPSL | 97659×100093 | available | available | available | available |
| GFDL-CM4 | GFDL | 110769×99690 | available | available | not available | not available |
| GFDL-ESM4 | GFDL | 110804×99690 | available | available | available | available |
| NorESM2-MM | NCC | 93221×99757 | not available | not available | not available | not available |
| NorESM2-LM | NCC | 93221×99757 | not available | not available | not available | not available |
| NorCPM1 | NCC | 54137×111951 | not available | not available | not available | not available |

* Full name of Institutions:
CCCma: Canadian Centre for Climate Modelling and Analysis (Canada)
CSIRO: Commonwealth Scientific and Industrial Research Organization and Bureau of Meteorology (Australia)
CMCC: Centro Euro-Mediterraneo per I Cambiamenti Climatici(Italy)
IPSL: L'Institut Pierre-Simon Laplace(France)
MPI: Max Planck Institute for Meteorology (Germany)
NCC: Norwegian Climate Centre (Norway)
NCAR: National Center for Atmospheric Research (US)
GFDL: Geophysical Fluid Dynamics Laboratory (US)

In this revision we clarify the reason for choosing a specific as - "The two prognostic variables dissolve inorganic carbon (DIC) and total alkalinity (TA) are the essential data needed to drive a regional oceanic carbon model. There is no time-varying observational products or reanalysis of DIC and TA that has an ideal 3-dimensinal coverage of the GoM. NCAR's CESM2-WACCM-FV2 solution was chosen to serve as the model boundary due to its relatively small bias in the carbonate variables in the GoM, relative high horizontal resolution in the GoM compared with other GCMs and its availability of nutrients and carbon variables (see Table A1 for more details)."

The philosophy of using GCM to drive the regional model is 1) unlike some non-volatile elements in the ocean such as nitrate, iron, silicate, etc, carbonate system variables undergoes evolution and long-term trends with climate change under elevated atmospheric $CO_2$ forcing. The purpose of setting up a carbon model is at least partly to reflect the evolution of these climate sensitive gases and carbonate ion concentration in the water body. Using a reanalysis product without progression defeats the modeling purpose. Secondly, the connections between physical and biogeochemical variables are essential for understanding the ocean carbon cycle. Carbonate variables are not isolated but an indiscerptible portion of the mass and energy transport in the ocean. One key shortcoming of reanalysis products is the separation of these connections/processes. Extensive gap-filling techniques were applied to generate the climatology/reanalysis products when there is limited observational data. The preference between reanalysis products and GCM product is, in essence, a debate between non-mechanism and mechanism-based estimation model. For example, the raw TA observations from 2000-01-01to 2019-12-31 at all water depth including coastal estuaries in the GoM is 4189 counts (data included NCEI Accession 0083633, 0117971, 0144622, 0154383, 0157025, 0157461, 0157619, 0188877, 0188878, 0188879, 0188976, 0188977, 0188978, 0189038, 0189291, 0189592, 0208096, 0209158, 0219960, 0231438, 0240147, 0240177, 0240205, 0240206, 0240314, 0240320, 0240322), this already includes all publicly accessible TA observation data in the GoM. Assuming each observation data point is 100% representative of a 5km×5km grid point at its depth at the given month, when using the 36 vertical depth layout in this study, the observational data can only fill 2959 distinct grid points in the model. To generate a fully covered reanalysis product, substantial efforts need to be made in extrapolation if possible. Thirdly, introduced artifacts from data extrapolation is a problem from reanalysis products, such artifacts are random/method-bound and not supported by biogeochemical processes, and using a non-mechanism-based forcing for setting up a mechanism-based model can make the model assessment and optimization unreliable. For instance, bias from over-upwelling or over-calcification may be mixed with extrapolation artifacts, this makes model diagnose and process optimization elusive. It should be noted that empirical relationships with temperature and salinity are widely employed for DIC and TA extrapolation and in climatology product preparation (e.g., Xue, et al. 2016).

The only two climatology products containing both DIC and TA for the global oceanic waters are "OceanSODA-ETHZ" (Gregor and Gruber, 2020) and "A global monthly climatology of total alkalinity (AT) and total dissolved inorganic carbon (DIC)" (Broullón et al., 2020a,b). However, none of them can provide three-dimensional DIC/TA data with time progression, with the former covering from 1985 through 2018 by taking advantage of satellite data yet only contain surface information and the latter being a 12-snapshot monthly product erased time component. We explain the reason for not using a reanalysis product concisely in the revised manuscript as well.

**RC 1:** *Figure 15*

*I recommend that the authors either remove figure 15 or reconsider the colour scheme. It is currently very difficult to make out where the model is positively or negatively biased. I can see that there is a large*

*bias near the coast, but elsewhere I cannot easily tell if there is a positive or negative bias. This could be fixed by using a diverging colour palette.*

**Response:** Thanks for the suggestion, we originally used a non-diverging color palette to distinguish the blank background and data points what has small bias. We have changed into a diverging color palette with the neutral color being the same as the blank background, and only included the regional model results. We added a statement that the blank background color does not indicate neutral bias in the figure caption to prevent confusion.

[Figure]

**Figure R1-1 (Figure 15 in new revision): Comparison of sea surface $pCO_2$ between regional ocean model products (Xue 2016, Gomez 2020, Chen 2019, This work), and underway sea surface $pCO_2$ measurements. A Positive $\Delta pCO_2$ indicates the product data overestimate sea surface $pCO_2$. A negative $\Delta pCO_2$ suggests the product data underestimate sea surface $pCO_2$. A neutral $\Delta pCO_2$ indicates the product data agree well with the observed sea surface $pCO_2$. The white spaces between the cruise lines indicate these regions do not have observational $pCO_2$ data, and do not indicate neutral bias.**

**RC 1:** *Figure 16.*

*This is an importamt and useful figure, in that shows the better performance of this model versus the others assessed. However, the authors should consider making some changes.*

*I would not expect the coarsely gridded CMIP6 models to be particularly good at resolving pCO2 in this region. It should be easy to outperform them, and so the comparison with CMIP6 should probably go into the supplementary materials.*

*The important comparison seems to be with the previous regional models. I therefore recommend redoing figures 15 and 16 to only include the regional studies. This is also more in line with the "Reassessment" aspect of the title. It is important the authors make totally clear how their study is improving on previous ones, and figures 15 and 16 weaken that aspect of the study.*

**Response:** Thanks for the suggestions for Figure 15 and 16, we modified the figures to only include the regional studies to be more in line with the "Reassessment". And the updated figure is also shown below.

[Figure]

**Figure R1-2 (Figure 16 in new revision): Comparison of sea surface $pCO_2$ among regional ocean model products (Xue 2016, Gomez 2020, Chen 2019, This work) at two buoy sites. Climatology at the two buoy locations of Gomez et al. (2020) is calculated by multiyear averaging from 2000-2014 model surface results. Climatology at the two buoy locations of Xue et al. (2016) is calculated by multiyear averaging from 2005 to 2010. Climatology at the two buoy locations of Chen et al. (2019) is calculated from their 12-monthly ML surface $pCO_2$ product (from 2002-07 to 2017-12). Buoy raw observations have a frequency of ~ 3 hours, and monthly averages are used to be compared with monthly model estimates. The p-value for each correlation coefficient is listed in the p-value table.**

**RC 1:** *Table 2*

*As above, the authors should consider restricting this to the regional models. Ideally, table 2 will only include the models shown in table 3. There is also a potential technical issue with the calculations for the CMIP6 models. Many of the coastal points will be outside the CMIP6 model domains. It is therefore unclear how they were handled. Were they extrapolated outside the domain? Given the heavy concentration of coastal points, it is possible the comparison between CMIP6 models and this study is not consistent..*

**Response:** We reorganized Table 2 to only include the models shown in Table 3. Thanks for pointing out the potential technical issue of directly comparing the observation with CMIP6 models. We have included a similar statement in the original manuscript that "(Line 48) However, their relatively coarse spatial resolution is likely not appropriate to be directly compared with field measurements". This is also a

reason to develop a regional model to better fill the gap between observations and models. In our first submission, we did not perform a such direct comparison between GCMs and observations, and it was added to provide more information upon request. Due to the coarsely resolved coastline, some coastal observation data points might be outside of the CMIP6 model domain. In such cases, we have to force the CMIP6 model to give its best estimation from the nearest data points of the nearest timespan. Indeed, observation data represent a much smaller geographical area over a limited time span. It is unfair for the CMIP6 models to be compared with the regional model or observations, in that they have different horizontal resolutions and time-frequency. However, the results are the best estimates CMIP6 monthly data products can provide. An extrapolation from the nearest data point is better than no estimation, if there were no other choice, global model products can still be a good source for making a such estimation. The product comparison is not consistent as limited by each product's capability, but it is consistent in that all products are allowed to give their best estimations.

*RC2: Review of "A Numerical Reassessment of the Gulf of Mexico Carbon System in Connection with the Mississippi River and Global Ocean"*

*This paper presents a high-resolution model of the Gulf of Mexico with a fully coupled marine carbon cycle included. It demonstrates that the model performs well and better than climate models in this region with respect to the variables evaluated. Furthermore, the model results are used to investigate trends and changes in the surface carbon variables and exchange with the atmosphere over the simulation period (20 years). The importance of lateral fluxes and riverine input to the region is also investigated. This paper makes a substantial contribution to the science of biogeochemical cycling in the Gulf of Mexico. This is the second time I review this paper and I think that my initial concerns have been comprehensively addressed and I therefore recommend the paper for publications with a few minor corrections.*

*The biggest request for change is that I think large parts of section 5.1, equations, figures, results fit better at the end of section 3 than in the discussion section. A small part where you discuss the differences between the models can be retained under 5.1.*

*I appreciate the fact that so few observations go into the climatology close to the boundary as shown in figure R1-1, but I think the units must be wrong in this figure: should it be micromoles/kg?*

*A couple of typos:*
*Line 16: should it be "generally interesting carbon …"?*
*Result, line 452: Indice should be index*

**Response**: Thanks for the careful review of this manuscript and the invaluable suggestions in both rounds of comments. We have re-examined the layout of section 5.1 and find the discussion content involving multiple global climatology products and global models necessary.  Thus we think it is reasonable to leave it as current section rather than being moved to the end of validation section (section 3).
The unit for Figure R1-1 should be *micromoles/kg*, sorry for this oversight. A corrected figure is attached below as Figure R2-1.